# *HMMR* acts in the PLK1-dependent spindle positioning pathway and supports neural development

**Marisa Connell[1†], Helen Chen[1†], Jihong Jiang[1], Chia-Wei Kuan[2], Abbas Fotovati[1], Tony LH Chu[1], Zhengcheng He[1], Tess C Lengyell[3], Huaibiao Li[4], Torsten Kroll[4], Amanda M Li[1], Daniel Goldowitz[3,5], Lucien Frappart[4], Aspasia Ploubidou[4], Millan S Patel[5], Linda M Pilarski[6], Elizabeth M Simpson[3,5], Philipp F Lange[2,7], Douglas W Allan[8], Christopher A Maxwell[1,7]\***

[1]Department of Paediatrics, University of British Columbia, Vancouver, Canada; [2]Department of Pathology and Laboratory Medicine, University of British Columbia, Vancouver, Canada; [3]Centre for Molecular Medicine and Therapeutics, University of British Columbia, Vancouver, Canada; [4]Leibniz Institute on Aging—Fritz Lipmann Institute, Beutenbergstrasse, Germany; [5]Department of Medical Genetics, University of British Columbia, Vancouver, Canada; [6]Cross Cancer Institute, Department of Oncology, University of Alberta, Edmonton, Canada; [7]Michael Cuccione Childhood Cancer Research Program, BC Children's Hospital, Vancouver, Canada; [8]Department of Cellular and Physiological Sciences, Life Sciences Centre, University of British Columbia, Vancouver, Canada

**\*For correspondence:** cmaxwell@bcchr.ubc.ca

[†]These authors contributed equally to this work

**Competing interests:** The authors declare that no competing interests exist.

**Abstract** Oriented cell division is one mechanism progenitor cells use during development and to maintain tissue homeostasis. Common to most cell types is the asymmetric establishment and regulation of cortical NuMA-dynein complexes that position the mitotic spindle. Here, we discover that HMMR acts at centrosomes in a PLK1-dependent pathway that locates active Ran and modulates the cortical localization of NuMA-dynein complexes to correct mispositioned spindles. This pathway was discovered through the creation and analysis of *Hmmr*-knockout mice, which suffer neonatal lethality with defective neural development and pleiotropic phenotypes in multiple tissues. HMMR over-expression in immortalized cancer cells induces phenotypes consistent with an increase in active Ran including defects in spindle orientation. These data identify an essential role for HMMR in the PLK1-dependent regulatory pathway that orients progenitor cell division and supports neural development.

DOI: https://doi.org/10.7554/eLife.28672.001

## Introduction

The development and homeostasis of the brain require that neural stem cells, termed neuroepithelial progenitors (NPs), balance expansion with differentiation (*Gönczy, 2008*; *Knoblich, 2008*). During neurogenesis, NPs primarily undergo divisions that produce the various types of cells required for proper brain development. During these divisions, the orientation of the axis of division and mitotic spindle position determines the placement of the progeny cells within the tissue and, as a consequence, imbalances in the spindle positioning pathway can lead to defects in brain morphology (*Lancaster and Knoblich, 2012*).

Mitotic spindle position is controlled by forces applied on astral microtubules by dynein molecular motor complexes, which are anchored at the cell cortex by complexes of nuclear mitotic apparatus

**eLife digest** As an embryo develops, its cells divide, grow and change into many different types of cells that eventually build our body. When cells divide, they first need to duplicate their genetic material. A structure called the spindle then distributes the two copies of the genetic information between the new cells. Cells must position their spindle precisely, and the way the spindle is oriented helps to determine what type of cell will develop. If the spindle fails to align properly, it can disrupt the development of specific tissues and organs and even lead to diseases such as cancer.

Numerous proteins help to position the spindle. For example, a protein called Ran-GTP ensures that motor proteins are anchored on opposite sides of the dividing cell, which tug on the spindle and position it between them. If the spindle gets pulled too closely to one side, a protein called PLK1 changes parts of the motor proteins to reduce the pulling force and to reposition the spindle towards the center.

Previous research has shown that non-motor proteins, such as a protein called HMMR are also part of the motor-protein complex. However, until now it was not known how HMMR was involved in repositioning the spindle during this process.

Now, Connell et al. have used mice that lacked HMMR to find out if it helps the cells in the brain to develop. The results show that without HMMR, very few mice were able to survive and many suffered from deformed and underdeveloped brains. In these mice, the orientation of the spindle changed and fewer cells of the correct type could be formed. Connell et al. then analyzed different types of cells grown in the laboratory to better understand how HMMR controls the position of the spindle. In all cases, HMMR formed a complex with Ran-GTP and was needed for the cells to orient their spindle correctly. When HMMR was absent, PLK1 could not work properly, and the spindle was positioned incorrectly.

This suggests that HMMR is essential for the spindle to align properly and is needed to help brain cells develop and become specialized. The next step will be to understand how HMMR, Ran and PLK1 work together during cell division. Studying mice that survive without HMMR offer an opportunity to examine how poorly aligned spindles affect their development.

DOI: https://doi.org/10.7554/eLife.28672.002

protein (NuMA)- G-protein signaling modulator 2 (GPSM2, aka LGN) (*Konno et al., 2008*; *Morin et al., 2007*; *Peyre et al., 2011*). In mammalian cultured cells, a gradient of Ran-GTP at chromosomes excludes LGN-NuMA complexes from lateral regions of the cell cortex to establish cortical asymmetry of these complexes and to position the spindle (*Kiyomitsu and Cheeseman, 2012*). The activity of dynein complexes anchored at polar regions of the cortex are regulated by the proximity of the spindle pole and its associated polo-like kinase 1 (PLK1) gradient (*Kiyomitsu and Cheeseman, 2012*) and complexes of dynein light chain 1 (DYNLL1)- hyaluronan mediated motility receptor (HMMR) (*Dunsch et al., 2012*). If one pole moves too close to the cortex, the centrosomal components of this pathway strip dynein from the cortex to reduce pulling forces and reposition the mitotic spindle. PLK1 activity, for example, stabilizes astral microtubules that are needed to strip dynein-dynactin complexes from the cortex (*Zhu et al., 2013*). HMMR is not proposed to be part of the NuMA-PLK1 spindle positioning pathway but rather to regulate cortical dynein through a complex containing CHICA (aka FAM83D) and DYNLL1 (*Dunsch et al., 2012*). However, HMMR and PLK1 activity are known to be interconnected during mitosis: during spindle assembly, PLK1 activity as measured by a kinetochore-localized, FRET-reporter construct is reduced following treatment of cells with siRNA targeting HMMR (*Chen et al., 2014*) and HMMR-threonine 703 is a putative substrate for PLK1 identified through mitotic phosphoproteome analysis (*Nousiainen et al., 2006*).

During cell division, HMMR acts downstream from Ran-GTP to localize targeting protein for XKLP2 (TPX2) and promote spindle assembly (*Groen et al., 2004*; *Joukov et al., 2006*; *Chen et al., 2014*; *Scrofani et al., 2015*). This action is reliant upon a carboxy-terminal basic leucine zipper (bZip) motif in HMMR, which targets it to the centrosome (*Maxwell et al., 2003*), and enables the activation of Aurora A by TPX2 (*Chen et al., 2014*) (*Scrofani et al., 2015*). Aurora A can directly phosphorylate PLK1 on Thr210 (*Macůrek et al., 2008*), which suggests that HMMR may impact the PLK1-dependent positioning pathway. Consistent with a potential role establishing the planar cell

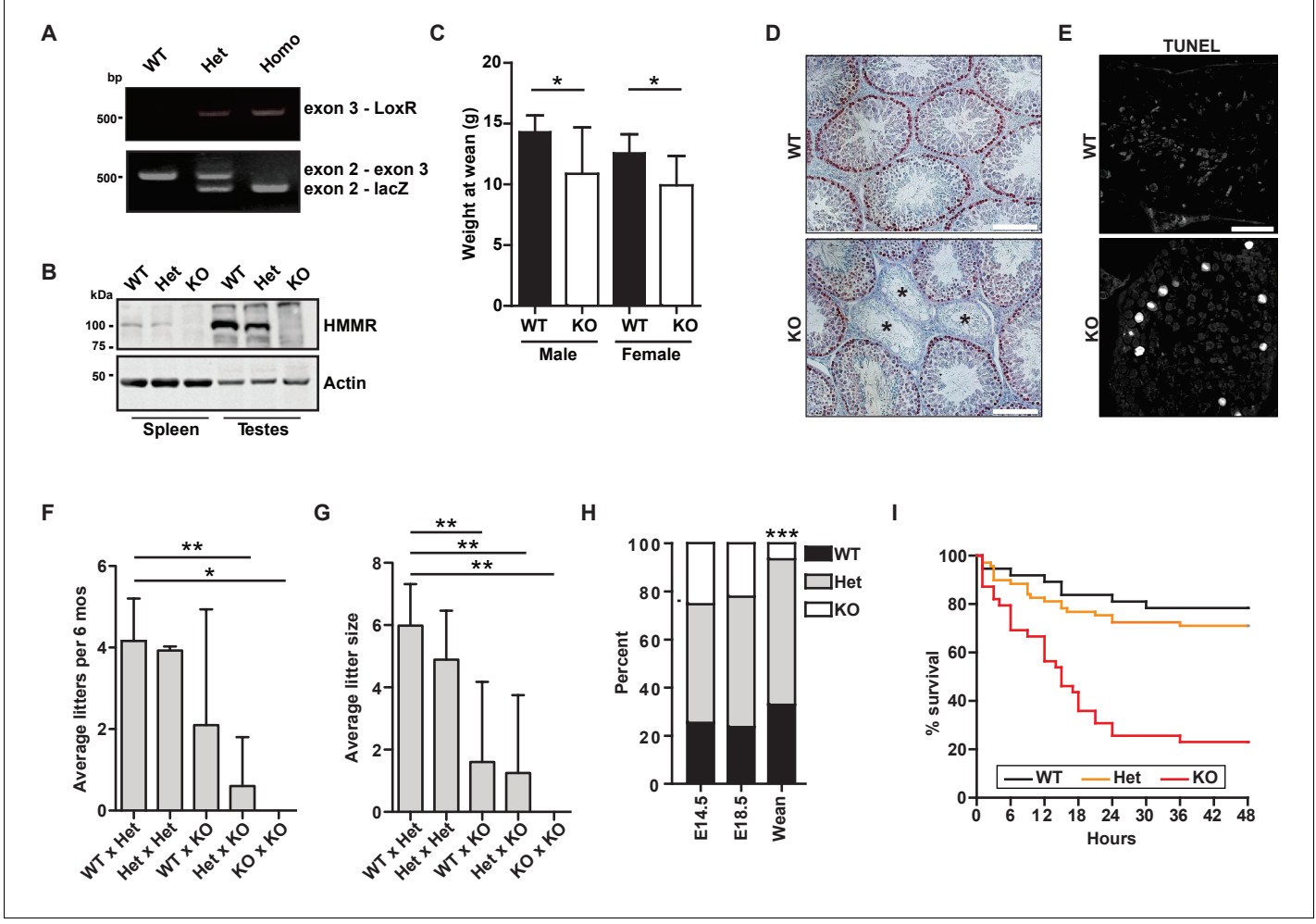

**Figure 1.** *Hmmr*[tm1a/tm1a] mice are smaller, exhibit fertility defects, and have decreased survival. (A) Genotyping PCR confirmed insertion of the targeting vector between exon 2 and exon 4 in *Hmmr*[tm1a/+] (Het) or *Hmmr*[tm1a/tm1a] (KO) but not in *Hmmr*[+/+] (WT) mice. (B) HMMR expression in tissues extracted from WT, Het, or KO mice. Actin served as a loading control. (C) Weight at wean for WT and KO mice. Data are represented as mean ±SD (*p=0.028 (males), p=0.022 (females); for males, n = 10 (WT), 3 (KO); for females, n = 12 (WT), 4 (KO)). (D) Defects in seminiferous tubules are present in a KO male (*, atrophic tubules) relative to age-matched WT mouse stained with H&E. Scale bars, 200 μm. (E) Apoptosis (TUNEL staining) in KO male seminiferous tubules relative to age-matched WT mouse. Scale bars, 100 μm. (F) Number of litters per 6 months breeding time for matings of WT, Het, and KO mice. (*p<0.05; **p<0.01; n = 11 matings (WT x Het), 2 (Het x Het), 4 (WT x KO), 3 (Het x KO), 2 (KO x KO)). (G) Pups per litter for matings of WT, Het, and KO mice (See *Figure 1F* for n values). (H) Percentage of WT, Het, and KO pups at E14.5, E18.5 and weaning (~21 days) (***p<0.001; n = 64 (E14.5), 49 (E18.5), 133 (wean)). (I) Survival analysis for WT, Het or KO neonates during the first 48 hr following birth (n = 34 (WT), 69 (Het), 36 (KO)).
DOI: https://doi.org/10.7554/eLife.28672.003

The following figure supplement is available for figure 1:

**Figure supplement 1.** Schematic of HMMR protein/gene, mouse models, and primer locations for detection of Hmmrtm1a targeting construct.
DOI: https://doi.org/10.7554/eLife.28672.004

division exhibited in neural progenitors, HMMR has been shown to regulate apicobasal polarity in mammary epithelia (*Maxwell et al., 2011*), while expression of a truncated *Hmmr* (1–317 of 794 aa, termed *Hmmr*[m/m]) impairs planar cell division in ovarian follicle cells (*Li et al., 2015*), and germ cells in the testes (*Li et al., 2016*).

HMMR is expressed in the developing nervous system (*Casini et al., 2010*) and proliferative regions of the adult mouse brain (*Lindwall et al., 2013*), Recently, HMMR has been shown to be required for anterior neural tube closure and morphogenesis in *Xenopus*, where HMMR reduction leads to the absences of ventricular lamina and increased intraocular distance, olfactory bulb size, and forebrain width (*Prager et al., 2017*). The N-terminal microtubule binding region in HMMR is

needed for neural tube morphogenesis in *Xenopus* (*Prager et al., 2017*) and the very terminal region is similar to that of Miranda (*Chang et al., 2011*), a regulator of asymmetric NP cell division in *Drosophila* (*Ikeshima-Kataoka et al., 1997*; *Shen et al., 1997*).

*Hmmr* mutant mice models are viable, including when central exons are targeted in *Hmmr*$^{-/-}$ mice (*Tolg et al., 2003*) and *Hmmr*$^{m/m}$ mice (*Li et al., 2015*), which result in the expression of truncated *Hmmr* transcript and protein (exons 1–7 or exons 1–10, respectively). Here, we studied the requirement of HMMR during oriented NP cell division and nervous system development through the creation of *Hmmr*-deficient mice with targeted disruption of *Hmmr* following exon 2. We find that HMMR is needed for neonatal survival and proper brain development. Our studies using cultured primary fibroblasts, directed differentiation of embryonic stem cells, and immortalized cancer cell lines, including neuroblastoma-like cells, uncovered a role for HMMR in the PLK1-dependent positioning pathway at mitotic spindle poles.

## Results

### *Hmmr*$^{tm1a/tm1a}$ neonates have reduced survival

We generated mice encoding a targeting construct following *Hmmr* exon 2, termed *Hmmr*$^{tm1a}$$_{(EUCOMM)Hmgu}$ (hereafter *Hmmr*$^{tm1a/tm1a}$, *Figure 1A*, *Figure 1—figure supplement 1A–B*). Western blot analysis of lysates from tissues known to have elevated levels of HMMR expression (spleen and testes) (*Line et al., 2002*) using antibodies targeting the N-terminal peptide in HMMR revealed that HMMR expression was completely lost in *Hmmr*$^{tm1a/tm1a}$ mice (*Figure 1B*). Adult *Hmmr*$^{tm1a/tm1a}$ mice were rare, and those *Hmmr*$^{tm1a/tm1a}$ mice that did survive were smaller than their wild-type (WT) littermates (*Figure 1C*). Similar to the phenotypes seen in *Hmmr*$^{m/m}$ mice attributed to misoriented germ cell divisions (*Li et al., 2016*), we observed atrophic seminiferous tubules and an increase in apoptosis in the testes as indicated by TUNEL staining in *Hmmr*$^{tm1a/tm1a}$ mice (*Figure 1D–E*). Additionally, *Hmmr*$^{tm1a/tm1a}$ mice were less fertile (fewer litters and fewer pups per litter) (*Figure 1F–G*). Few adult *Hmmr*$^{tm1a/tm1a}$ mice survived despite no evidence of embryonic lethality at E14.5 and E18.5 (*Figure 1H*). To identify when *Hmmr*$^{tm1a/tm1a}$ mice were dying, we monitored neonates for 2 days following birth. 12.5% of *Hmmr*$^{tm1a/tm1a}$ neonates were found dead within 3 hr of birth and 76.9% were found dead within the first 48 hr after birth (*Figure 1I*).

### *Hmmr*$^{tm1a/tm1a}$ mice display defects in brain structure, neural progenitor division, and differentiation

Necropsy samples from *Hmmr*$^{tm1a/tm1a}$ neonates (P0-1) demonstrated morphological defects in the brain, including defects in overall structure and size (*Figure 2A*). In multiple matching sections taken from WT or *Hmmr*$^{tm1a/tm1a}$ neonatal brains, we measured the area of the cerebrum and ventricles. We found large variation in the size of *Hmmr*$^{tm1a/tm1a}$ neonatal brains with three of the nine measuring two standard deviations smaller (microcephaly) than the mean brain size for age-matched (P0-1) WT littermates (*Figure 2B*). In addition, three of nine *Hmmr*$^{tm1a/tm1a}$ neonatal brains measured two standard deviations larger (megalencephaly) than the mean brain size for age-matched (P0-1) WT littermates, but these larger areas correlated with increases in the ventricle area in two of these three *Hmmr*$^{tm1a/tm1a}$ brains (*Figure 2C*). The putative reduced cortical area identified in *Hmmr*$^{tm1a/tm1a}$ relative to WT brains was not accompanied by increased levels of apoptosis, as measured by TUNEL staining, at either E18.5 or P0-1 (*Figure 2—figure supplement 1A*).

A putative reduction in cortical area and the potential manifestation of enlarged ventricles (hydrocephalus) in *Hmmr*$^{tm1a/tm1a}$ mice is consistent with phenotypes associated with defects in the orientation of NP cell division (*Noatynska et al., 2012*; *Kielar et al., 2014*). As HMMR is known to regulate the cell division axis in immortalized cancer cells (*Dunsch et al., 2012*), we next examined the localization of HMMR and the axis of NP cell division during neurogenesis in E14.5 embryos. In WT embryos, HMMR localized to spindle microtubules in dividing NP cells lining the ventricle surface (*Figure 2D*). The angle of NP cell divisions measured during anaphase was strongly biased to be planar with the ventricle surface (<30 degrees off the axis of centrosomes lining the surface) with an average spindle angle of 18.0 ± 19.5 degrees (*Figure 2E–F*). In WT brains, 60.6% or 82.9% of divisions were oriented within 15 degrees or 30 degrees of the ventricular surface, respectively. In *Hmmr*$^{tm1a/tm1a}$ embryos, however, HMMR was absent from mitotic spindles (*Figure 2D*) and the axis

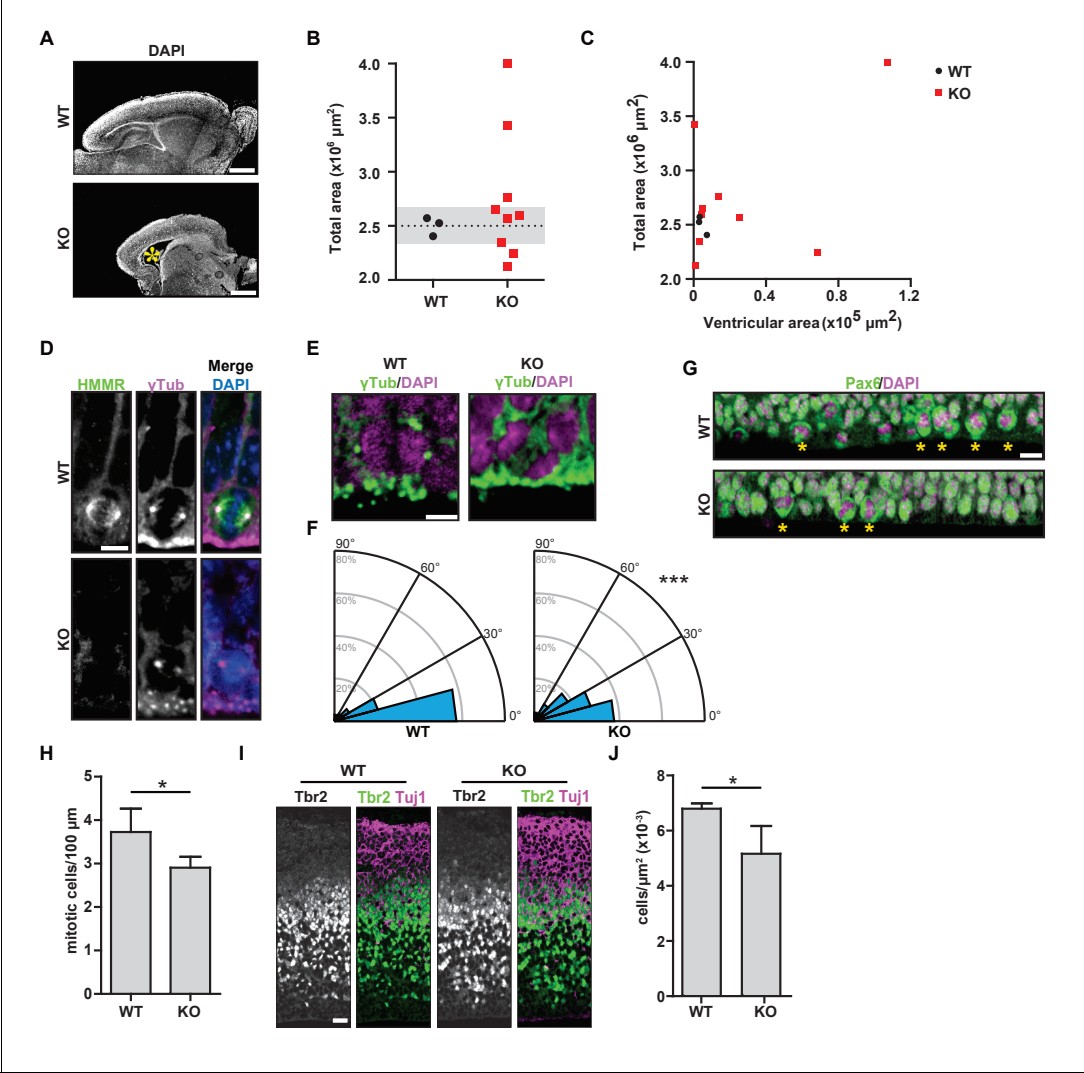

**Figure 2.** *Hmmr^{tm1a/tm1a}* mice display defects in brain structure, neural progenitor division, and differentiation. (A) *Hmmr^{tm1a/tm1a}* (KO) neonate brains showed abnormalities compared to *Hmmr^{+/+}* (WT) brains (*, enlarged ventricles). Scale bars, 500 μm. (B) Quantification of brain size in WT and KO neonate (P0-1) brains. Dotted line indicates average of WT brains. The grey region indicates two standard deviations from the WT mean (n = 3 (WT), 9 (KO), p=0.512). (C) Correlation between brain size and ventricular area in WT and E14.5 P0-1 brains. (D) HMMR localization to the mitotic spindle (γ-tubulin) in neuroepithelial progenitor (NP) cells of WT and KO E14.5 brains. Scale bar, 10 μm. (E) Mitotic NP cells labeled with γ-tubulin (centrosomes) and DAPI in WT and KO E14.5 brains. Scale bar, 5 μm. (F) Division angle for NP cells in WT or KO E14.5 embryos (***p<0.001; n = 94 (WT), 96 (KO)). (G) Mitotic rate in Pax6^+ NP cells in WT and KO E14.5 brains. Stars indicate dividing cells. Scale bar, 10 μm. (H) Quantification of mitotic rate in WT and KO E14.5 brains. Data are represented as mean ±SD (*p=0.018; n = 4(WT), 5(KO)). (I) Tbr2 localization in WT and KO E14.5 brains. Scale bars, 20 μm. (J) Quantification of Tbr2^+ cells in E14.5 brains. Data are represented as mean ±SD (*p=0.019; n = 4 (E14.5, WT), 5 (E14.5, KO)).

DOI: https://doi.org/10.7554/eLife.28672.005

The following figure supplement is available for figure 2:

**Figure supplement 1.** *Hmmr^{tm1a/tm1a}* mice do not display increased levels of apoptosis, defects in polarity, or centrosome amplification.

DOI: https://doi.org/10.7554/eLife.28672.006

for division was altered with an average angle of 24.8 ± 19.5 degrees (p<0.05; unpaired t-test) with only 38.5% or 66.6% of divisions oriented within 15 degrees or 30 degrees of the ventricular surface, respectively ($\chi^2$ test, p<0.001) (*Figure 2E–F*). The degree of misorientation observed in *Hmmr^{tm1a/tm1a}* NP cells compares to that observed in *LGN*-mutant NP cells, which have 37.1% or 60% of divisions oriented within 15 degrees or 30 degrees of the ventricular surface, respectively (*Konno et al., 2008*). We also observed a significant reduction in the number of mitotic Pax6^+ radial glial cells at E14.5 in the brains of *Hmmr^{tm1a/tm1a}* embryos (*Figure 2G–H*) and a consistent, significant reduction

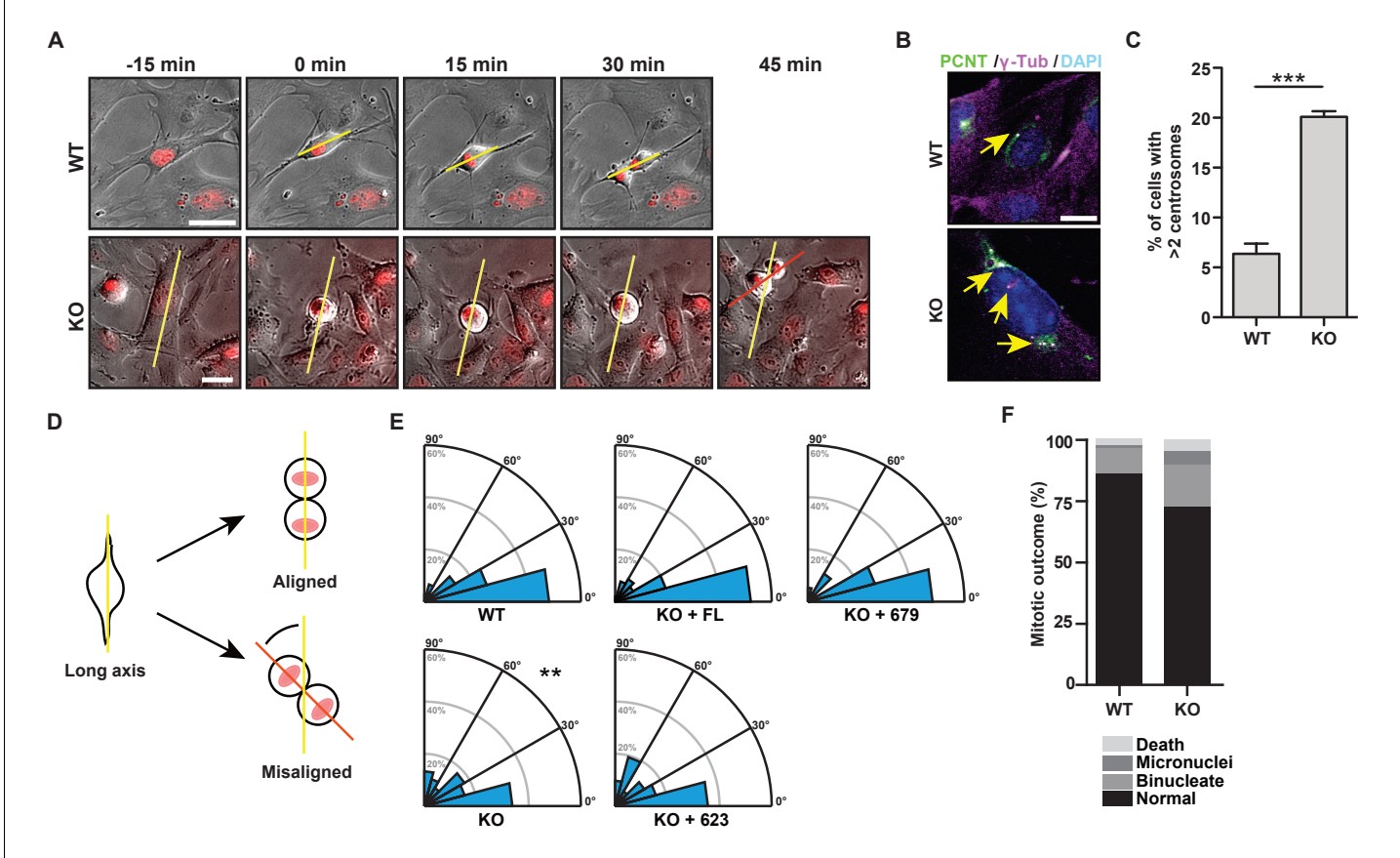

**Figure 3.** The bZip domain of HMMR is required for spindle orientation. (**A**) Cell division in mouse embryonic fibroblasts (MEFs) isolated from *Hmmr*<sup>+/+</sup> (WT) or *Hmmr*<sup>tm1a/tm1a</sup> (KO) embryos. DNA is counterstained with Hoechst (red). Yellow line indicates long axis of interphase cell and red line indicates cell division axis. Scale bars, 20 µm. (**B**) Centrosome amplification in WT or KO MEFs indicated by colocalization of pericentrin and γ-tubulin. Arrow indicates centrosome. (**C**) Quantification of centrosome amplification in WT or KO MEFs. Amplification was counted as >2 centrosomes per cell as indicated by colocalization of pericentrin and γ-tubulin. Data are represented as mean ±SD (**p=0.0002; 3 replicates of ≥100 cells). (**D**) Schematic of spindle angle measurement in MEFs. (**E**) Spindle angle in WT or KO MEFs. KO MEFs were transduced with HMMR constructs truncated at the indicated amino acid. WT and KO average spindle angle are significantly different (**p<0.01; n = 117 (WT), 114 (KO), 50 (KO + FL), 56 (KO + 679), and 31 (KO + 623)). (**F**) Mitotic outcomes in WT or KO MEFs (n = 111 (WT) and 94 (KO)).

DOI: https://doi.org/10.7554/eLife.28672.007

The following figure supplement is available for figure 3:

**Figure supplement 1.** *Hmmr*<sup>tm1a/tm1a</sup> MEFs are similarly elongated to Hmmr+/+MEFs.

DOI: https://doi.org/10.7554/eLife.28672.008

in Tbr2<sup>+</sup> intermediate progenitor cells in *Hmmr*<sup>tm1a/tm1a</sup> brains (*Figure 2I–J*). However, we observed no significant difference between WT and *Hmmr*<sup>tm1a/tm1a</sup> brains in the levels of TUNEL-positive cells in the cortex, the localization of Par3 and ZO-1 in cells lining the ventricles, or the levels of centrosome amplification in dividing NP cells (*Figure 2—figure supplement 1A–C*). These data support the conclusion that the reduced cortical areas observed in *Hmmr*<sup>tm1a/tm1a</sup> neonatal mice, which are less able to successfully transition to extrauterine life, result from the reduced production of intermediate progenitor cells that correspond with impaired planar spindle orientation during NP cell division and an overall decreased mitotic rate of Pax6<sup>+</sup> radial glial cells.

## Mitotic spindle orientation requires the centrosome targeting bZip motif in HMMR

To better understand the mechanisms through which HMMR controls spindle orientation, we isolated mouse embryonic fibroblasts (MEFs) from WT or *Hmmr*<sup>tm1a/tm1a</sup> mice and imaged these cells through mitosis (*Figure 3A*). *Hmmr*<sup>tm1a/tm1a</sup> MEFs contained significantly higher levels of centrosome

amplification (defined as cells containing greater than two centrosomes, which were quantified using colocalization of pericentrin and γ–tubulin) (*Figure 3B–C*), and were less able to orient the cell division axis, measured at anaphase relative to their long cell axis in interphase (*Figure 3A,D–E*). In addition, *Hmmr*$^{tm1a/tm1a}$ MEF progeny cells were more likely to display indications of genome instability, including increases in the frequency of mitotic death, binucleated progeny and micronuclei (*Figure 3F*). To ensure that the observed spindle orientation defects were not due to changes in cell shape that may accompany the loss of HMMR, we confirmed that the width to long axis ratio of the cell did not differ between WT or *Hmmr*$^{tm1a/tm1a}$ MEFs (*Figure 3—figure supplement 1A*). Thus, *Hmmr*$^{tm1a/tm1a}$ MEFs are less able to correctly orient the axis of cell division, which is similar to the defective spindle orientation that was observed following the treatment of HeLa cells with siRNA targeting HMMR (*Dunsch et al., 2012*).

To uncover the critical domain in HMMR needed to establish the cell division axis, we transduced GFP-HMMR (human) constructs into *Hmmr*$^{tm1a/tm1a}$ MEFs. We used truncations of the bZip domain, which is required to target HMMR to the centrosome (*Maxwell et al., 2003*), including GFP-HMMR$^{FL}$ (full-length, FL), HMMR$^{1-679}$ (truncation following the b-Zip domain), and HMMR$^{1-623}$ (truncation lacking the b-Zip domain). The transduction of HMMR$^{FL}$ or HMMR$^{1-679}$, gene products that are known to locate at centrosomes (*Figure 3—figure supplement 1B*)(*Maxwell et al., 2003*), recovered the axis of cell division (*Figure 3E*). However, the expression of HMMR$^{1-623}$ failed to rescue the axis of cell division (*Figure 3E*), implicating the centrosome targeting bZip motif (amino acids 624–679) as an essential domain for the orientation of cell division.

## Fixed spindle position and asymmetric cortical NuMA-dynein localization requires HMMR

As the centrosome targeting domain was critical to HMMR function in spindle positioning, we next used time-lapse imaging of HeLa cells that stably expressed DHC-GFP and were previously treated with either scrambled control siRNA or siRNA targeting HMMR to determine whether the centrosome-localized, PLK1-dependent positioning pathway, which regulates the localization and activity of dynein complexes anchored at polar regions of the cortex (*Kiyomitsu and Cheeseman, 2012*), was functional in HMMR-silenced cells. As multipolar spindles occur at an elevated frequency in HMMR-silenced cells (*Maxwell et al., 2005*), we restricted our analysis to mitotic cells with phenotypically normal bipolar spindles. Immediately prior to anaphase in control-treated cells with an off-center mitotic spindle, DHC-GFP was asymmetrically localized to the far polar region of the cortex and absent from the cortex near the proximal pole leading to the correction of spindle position (*Figure 4A*, *Video 1*) as described previously (*Kiyomitsu and Cheeseman, 2012*). In HMMR-silenced cells with an off-center mitotic spindle, however, DHC-GFP remained on the polar cortex and its retention on the cortex predicted the direction of spindle rotation (*Figure 4A*, *Video 2*). We also measured the intensity of DHC-GFP recruited to the cortex relative to the cytoplasmic intensity and determined that overall DHC-GFP recruitment to crescents was not significantly different between scrambled control-treated and HMMR-silenced cells (*Figure 4—figure supplement 1A*). In addition, the spindle was maintained in an off-centered position in HMMR-silenced cells as determined by the ratio of the distance between the cortex and the location of chromosomes in anaphase cells (*Figure 4B*). A prolonged period of spindle rotation was frequently observed in HMMR-silenced HeLa cells expressing DHC-GFP (*Figure 4C*) and, as seen in separate experiments, in HMMR-silenced HeLa cells expressing mCherry histone H2B, eGFP-TUBA (*Figure 4—figure supplement 1B–C*; *Videos 3–4*).

We next examined the localization of the dynein anchoring protein NuMA in fixed HMMR-silenced cells by immunofluorescence. When the fluorescence intensity of DHC-GFP or NuMA was measured along the cortex in control-treated cells with an off-center spindle, the ratio of intensities along the far polar cortex was elevated compared to that along the proximal polar cortex (*Figure 4D–E*), as expected. However, NuMA was retained at the proximal polar cortex and the consequent ratio of intensities was significantly dampened in HMMR-silenced mitotic cells with an off-center metaphase spindle (*Figure 4D–E*). Yet, when the metaphase spindle was centered, the asymmetric cortical localization of NuMA and DHC-GFP was indistinguishable in control-treated and HMMR-silenced cells (*Figure 4F–G*). That is, the ratio of intensities along the polar cortex relative to the midzone cortex was elevated at the polar cortex for both NuMA and DHC-GFP, as reported (*Kiyomitsu and Cheeseman, 2012*). Our data indicate HMMR is dispensable for the loss of NuMA

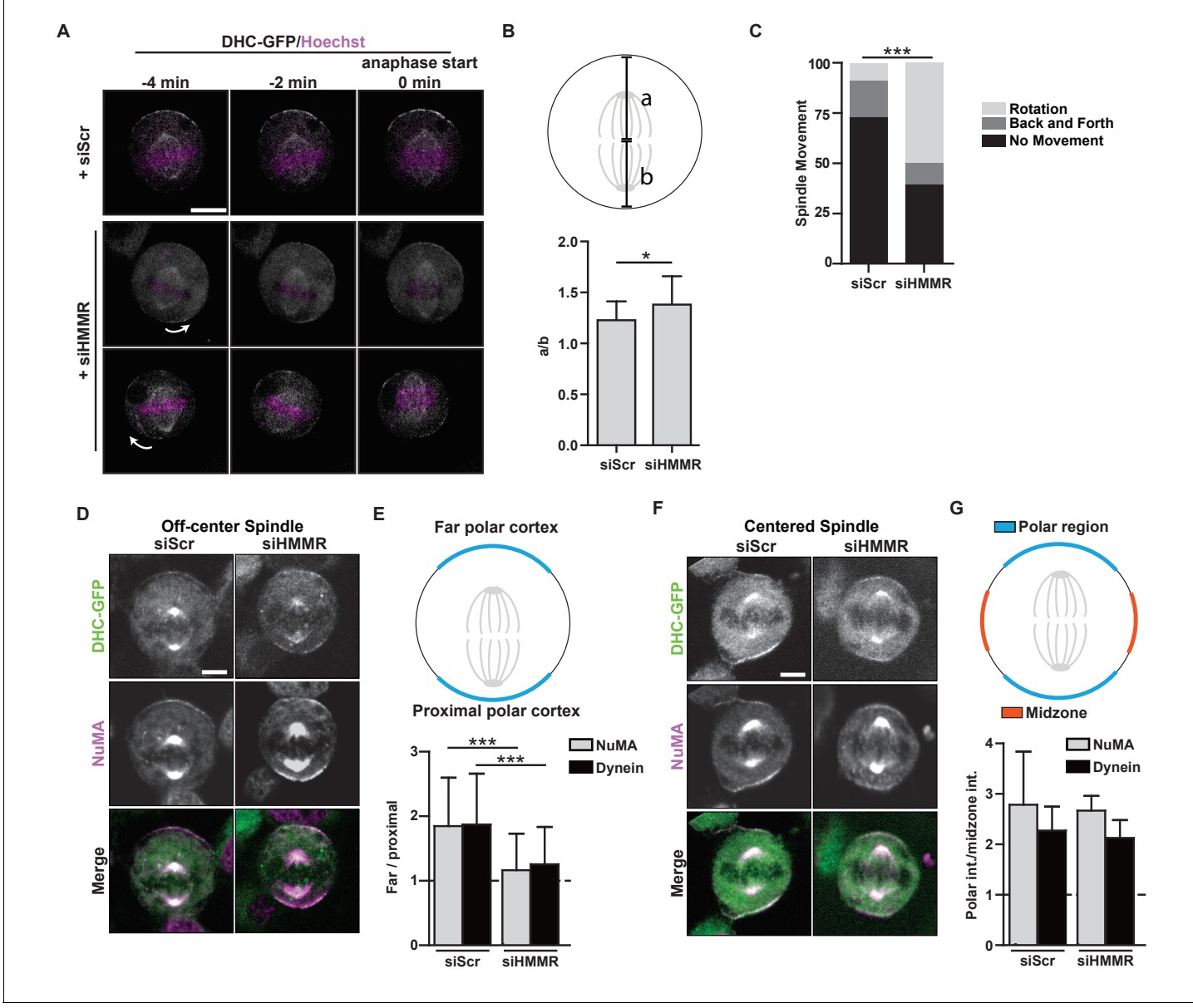

**Figure 4.** HMMR regulates cortical NuMA/Dynein localization. (**A**) Localization of Dynein heavy chain tagged with GFP (DHC-GFP) stably expressed in HeLa cells treated with scrambled (siScr) or HMMR (siHMMR) siRNA using time-lapse imaging (arrows indicate predicted direction of movement). Scale bar, 10 μm. (**B**) Quantification of off-center spindles at anaphase start in cells treated with siScr or siHMMR (n = 25 (siScr), 26 (siHMMR)). (**C**) Quantitation of spindle movement in cells treated with siScr or siHMMR. Back and forth movement is exemplified by *Figure 5A* (top panels) and rotation is exemplified by *Figure 5A* (middle and bottom panels) (p<0.001; n = 22 (siScr), 28 (siHMMR)). (**D**) Localization of NuMA and DHC-GFP in HeLa cells transfected with siScr or siHMMR with mispositioned spindles. Scale bar, 5 μm. (**E**) Quantification of NuMA and DHC-GFP intensities at the farthest polar cortex compared to the proximal polar cortex in HeLa cells transfected with siScr or siHMMR. Cells are pooled from experiments in panels 1E,F. (***p<0.0001; n = 39 (untreated), 32 (siScr), 53 (siHMMR)). (**F**) Localization of DHC-GFP and NuMA in HeLa cells transfected with siScr or siHMMR. Scale bar, 5 μm. (**G**) Quantification of NuMA and DHC-GFP at the poles compared to the midzone in HeLa cells transfected with siScr or siHMMR. All data are represented as mean ±SD (p<0.001; 3 replicates of ≥25 cells).

DOI: https://doi.org/10.7554/eLife.28672.009

The following figure supplement is available for figure 4:

**Figure supplement 1.** HMMR regulates spindle movement.

DOI: https://doi.org/10.7554/eLife.28672.010

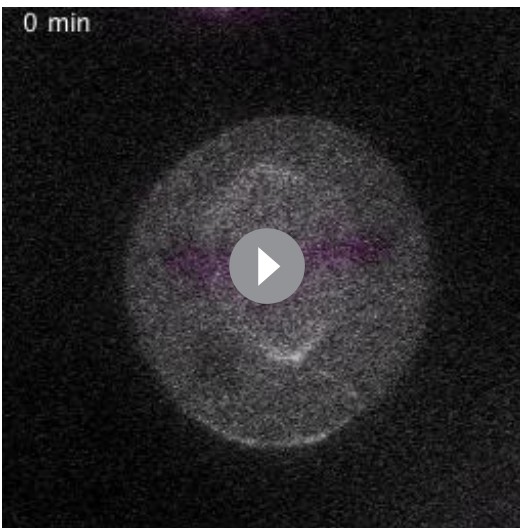

**Video 1.** Dynein localization in HeLa DHC-GFP cells treated with scrambled siRNA that possess off-center spindles. Images were captured every 1 min.
DOI: https://doi.org/10.7554/eLife.28672.011

and DHC-GFP at the midzone cortical region, which is established by Ran-GTP gradient generated at chromosomes (*Kiyomitsu and Cheeseman, 2012*), but HMMR is needed for a centrosome-localized positioning pathway that strips NuMA and DHC-GFP when the pole is brought proximal to the cortex.

## HMMR enables the PLK1-dependent spindle pole positioning pathway

We identified a requirement for HMMR in the removal of cortical NuMA-DHC-GFP, which is consistent with a prior report (*Dunsch et al., 2012*). However, we observed spindle rotation in HMMR-silenced cells rather than fixed and misoriented spindles (*Dunsch et al., 2012*). It was postulated that HMMR-DYNLL1 complexes at the centrosome functioned independent of the PLK1 intrinsic positioning pathway (*Dunsch et al., 2012*). That is, spindle pole-localized HMMR may create a local increase of DYNLL1 at the spindle/spindle poles that binds to dynein in competition with dynactin and displaces dynein from the cortex (*Dunsch et al., 2012*); however, Dyn2 (yeast DYNLL1 ortholog) promotes, rather than impedes, the incorporation of dynactin into the dynein motor complex (*Stuchell-Brereton et al., 2011*). So, we examined whether the loss of HMMR altered the composition of the dynein motor complex. HeLa cells expressing DHC-GFP were treated with control siRNA or siRNA targeting HMMR and DHC complexes were immunoprecipitated with antibodies recognizing GFP. In HMMR-silenced lysates, relative to control-treated lysates, the abundance of subunits for dynein (DHC, dynein intermediate chain (DIC), and DYNLL1) and dynactin (p150[glued]) were unchanged (*Figure 5A*). While the amount of DHC-GFP precipitated and the levels of DIC co-precipitated with DHC-GFP remained unaffected, the level of co-precipitated DYNLL1 was reduced in HMMR-silenced lysates (*Figure 5A*).

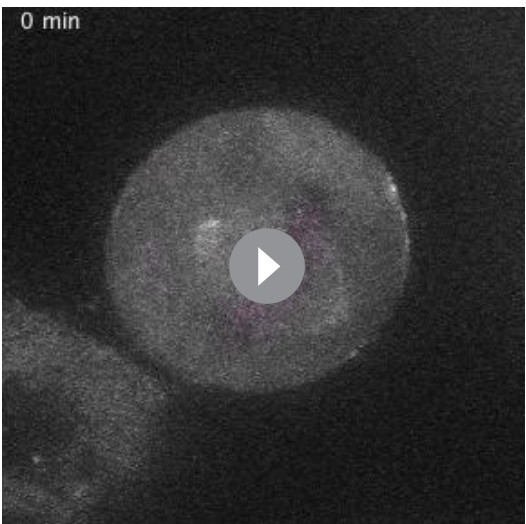

**Video 2.** Dynein localization in HeLa DHC-GFP cells treated with siRNA targeting HMMR that possess off-center spindles. Images were captured every 1 min.
DOI: https://doi.org/10.7554/eLife.28672.012

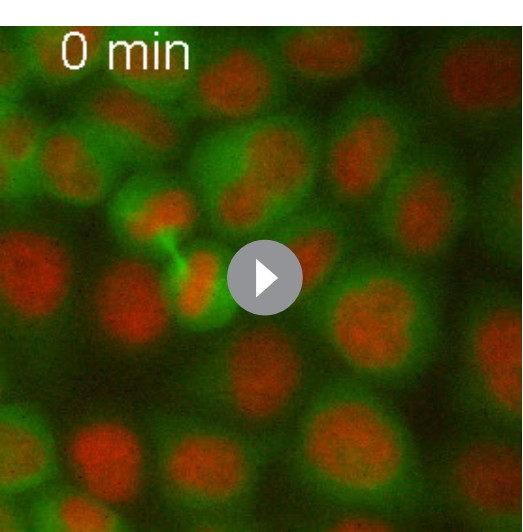

**Video 3.** Spindle movement in HeLa cells expressing mCherry-histone H2B, eGFP-TUBA treated with scrambled siRNA. Images were captured every 15 min.
DOI: https://doi.org/10.7554/eLife.28672.013

Moreover, a corresponding reduction in p150[glued] co-precipitated with DHC-GFP was also observed in HMMR-silenced lysates (*Figure 5A*). We also noted that both FLAG-DYNLL1 and CHICA were retained at spindle poles, although absent from spindle fibers, in HMMR-silenced cells (*Figure 5B–C*). Therefore, HMMR appears dispensable for the spindle pole localization of CHICA and DYNLL1 and HMMR promotes, rather than restricts, the incorporation of p150[glued] into DHC complexes. As our data do not support the proposed PLK1-independent role for HMMR in spindle positioning, we investigated a putative role for HMMR in the PLK1-dependent position-ing pathway.

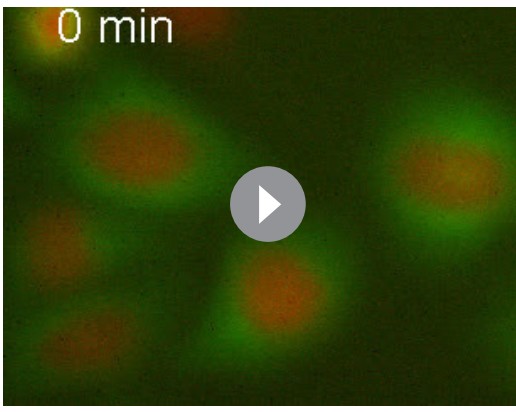

**Video 4.** Spindle movement in HeLa cells expressing mCherry-histone H2B, eGFP-TUBA treated with siRNA targeting HMMR. Images were captured every 15 min. DOI: https://doi.org/10.7554/eLife.28672.014

PLK1 activity at centrosomes and kinetochores enables the removal of LGN-NuMA-DHC com-plexes from the cortex (*Kiyomitsu and Cheese-man, 2012*; *Zhu et al., 2013*; *Tame et al., 2016*). As PLK1 activity is reduced at kinetochores in HMMR-silenced cells (*Chen et al., 2014*), we measured the levels of phosphorylated-PLK1, the active form of the kinase, by immunofluorescence in HMMR-silenced and control-treated mitotic cells. In HMMR-silenced mitotic cells, we observed a significant reduction in the fluorescence intensity of pPLK1 at spindle poles (*Figure 5D–E*). Consistent with a putative reduction in PLK1 activity at spin-dle poles, we observed a significant decrease in the number of EB1-positive microtubule plus ends contacting the cortex in HMMR-silenced or BI2536-treated mitotic cells (*Figure 5F–G*). As these data suggest HMMR enables PLK1-dependent processes, we tested if the concurrent inhibition of HMMR and PLK1 significantly altered the spindle positioning pathway when compared to the inhibi-tion of either alone. We focused on the localization of NuMA to the cortical regions most proximal to the spindle pole in control treated, HMMR-silenced, BI2536-treated, or dual inhibited metaphase cells with off-center spindles (*Figure 5H*). In control-treated cells, NuMA was uniformly absent from the cortex, while NuMA was largely retained in HMMR-silenced cells with the exception of a discreet loss in the inner region directly proximal to the pole (*Figure 5H–I*). In mitotic cells treated with BI2536, the localization of NuMA mirrored that observed in HMMR-silenced cells and was not addi-tively altered in dual inhibited cells (*Figure 5H–I*) consistent with a requirement for PLK1 and HMMR in a shared positioning pathway. Thus, our data supports the conclusion that astral microtubule den-sity is dampened in HMMR-silenced mitotic cells in a PLK1-dependent manner, which disturbs the spindle pole-localized positioning pathway and results in spindle rotation due to the retention of NuMA-DHC complexes at the proximal cortex.

## HMMR interacts with RanBP2 and regulates the centrosome localization of Ran

As PLK1 phosphorylates HMMR at threonine 703 (pHMMR) (*Nousiainen et al., 2006*), we used mass spectrometry to identify proteins that are co-precipitated with antibodies targeting pHMMR as a means to discover putative pathways through which centrosome-localized HMMR may regulate spin-dle position. In pHMMR immunoprecipitates from mitotic or G2-phase synchronized HeLa cell lysates, we identified a number of actin-, myosin-, and dynein-binding proteins (*Figure 6—figure supplement 1*; *Figure 6—source data 1*), including known interactors FAM83D/CHICA and DYNLL1 (*Dunsch et al., 2012*). We filtered out proteins that were also co-precipitated with antibodies target-ing a non-phosphorylated N-terminal peptide in HMMR (unpublished results) and found that pro-teins related to small GTPases, such as ARHGAP17, RACGAP1, and RanBP2, were uniquely co-precipitated with pHMMR antibodies (*Figure 6A*). As Ran regulates cortical LGN-NuMA-dynein localization during mitosis (*Kiyomitsu and Cheeseman, 2012*), we focused on the putative pHMMR-RanBP2 interaction and confirmed this interaction by western blot analysis (*Figure 6B*).

RanBP2 binds specifically to Ran-GTP (*Vetter et al., 1999*), so we postulated that HMMR may affect Ran-GTP levels or localization. To test this postulate, we first measured the levels of Ran-GTP

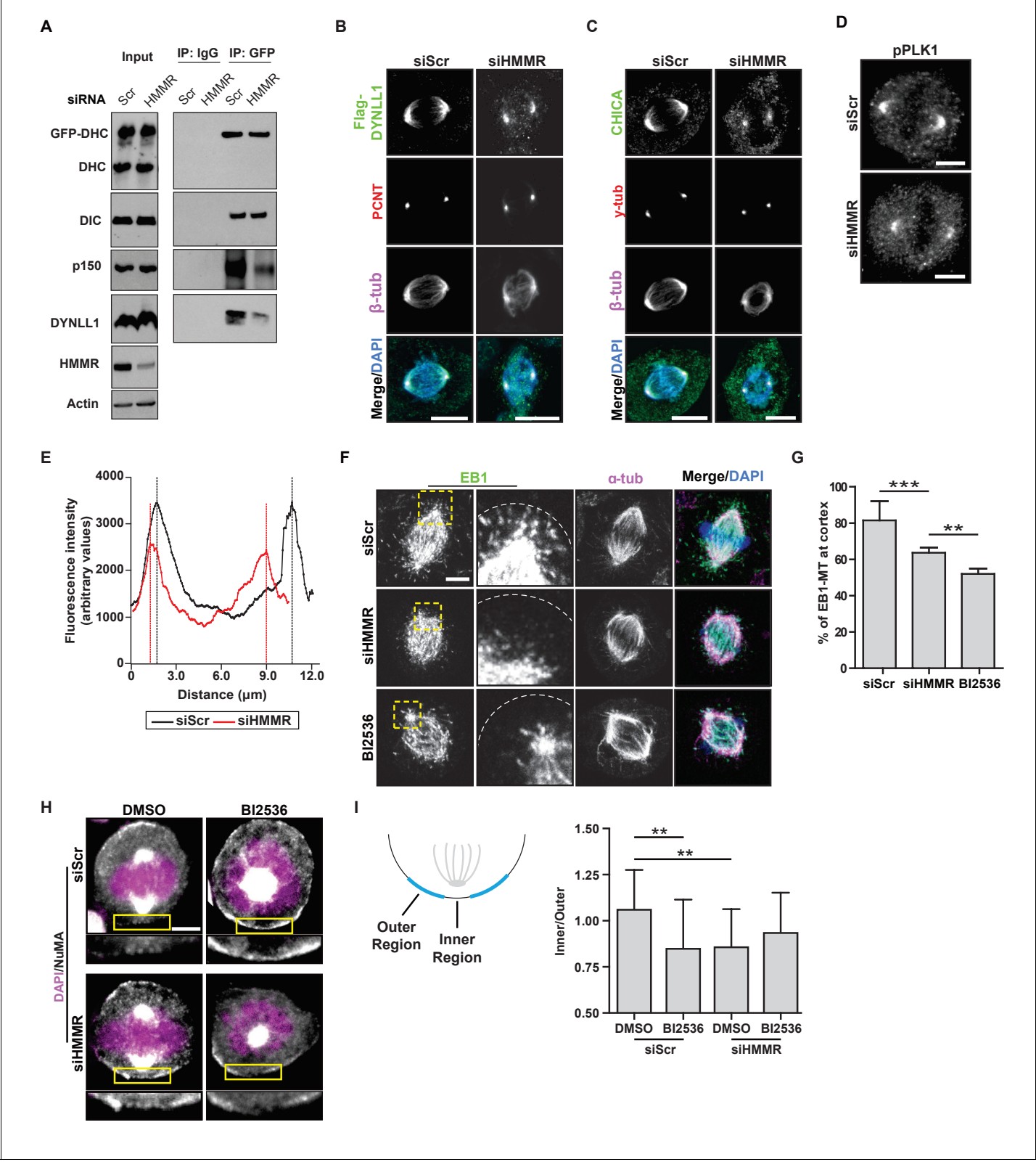

**Figure 5.** HMMR enables the PLK1-dependent spindle pole positioning pathway. (**A**) Western blot analysis of mitotic HeLa extracts stably expressing DHC-GFP treated with scrambled (siScr) siRNA or siRNA targeting HMMR (siHMMR) subjected to immunoprecipitation with GFP antibody or control IgG and blotted with the indicated antibodies, including GFP (GFP-DHC), dynein intermediate chain (DIC), p150[glued], DYNLL1, HMMR or Actin (four replicates). (**B**) Localization of Flag-Dynein light chain (DYNLL1) in HeLa cells treated with siScr or siHMMR (three replicates). Scale bars, 10 μm. (**C**)

*Figure 5 continued on next page*

Figure 5 continued

Localization of CHICA in cells treated with siScr and siHMMR (three replicates). Scale bars, 10 μm. (D) Localization of phospho-PLK1 (pPLK1) in HeLa cells expressing siScr or siHMMR. Scale bars, 5 μm. (E) Quantification of pPLK1 intensity from pole to pole. Data represents mean (50 cells from three experiments). (F) Astral microtubules contact the cortex in cells treated with siScr, siHMMR, or BI2536 (Plk1 inhibitor). Yellow box indicates EB1 inset and white dotted line indicates region of quantification. Scale bar, 5 μm. (G) Quantification of EB1 at the cortex in cells treated with siScr, siHMMR, or BI2536. Data are represented as mean ±SD (***p<0.001; 2 replicates of >40 cells per treatment). (H) NuMA localization in HeLa cells treated with siScr or siHMMR and DMSO or BI2536 (PLK1 inhibitor, 20 nM). Yellow box indicates magnified region. Scale bar, 5 μm. (I) Quantification of NuMA localization at the cortex near to the centrosome at the region nearest the centrosome compared to those farther away. Data are represented as mean ±SD (**p<0.01; n = 29 (siScr + DMSO), 39 (siScr + BI2536), 25 (siHMMR + DMSO), 39 (siHMMR + BI2536) from three experiments).

DOI: https://doi.org/10.7554/eLife.28672.015

in scrambled siRNA-treated or HMMR-silenced HeLa cell lysates with a commercially available assay that pulls-down the active form of Ran (Ran-GTP). In control experiments, scrambled siRNA-treated HeLa cell lysates were treated with either non-hydrolysable GTP (GTPγS) or with GDP to switch all Ran to Ran-GTP or Ran-GDP, respectively, prior to immunoprecipitation. As expected, the level of co-precipitated Ran was significantly augmented in cell lysates treated with GTPγS and lost in cell lysates treated with GDP (*Figure 6C*). Having verified our control conditions, we then compared the levels of Ran-GTP in control-treated and HMMR-silenced cell lysates. In HMMR-silenced cell lysates, the level of Ran-GTP precipitated was slightly reduced relative to scrambled control-treated cell lysates (*Figure 6C*). When we confirmed the knockdown efficacy of HMMR in these experiments, we noted that HMMR was also co-precipitated when lysates were pretreated with GTPγS suggesting HMMR may interact specifically with Ran-GTP, which is greatly increased by GTPγS treatment. To titrate a putative HMMR-Ran-GTP interaction, we precipitated HMMR from mitotic cell lysates previously treated with increasing amounts of GTPγS. While the levels of precipitated HMMR remained constant, we found that the level of Ran co-precipitated with HMMR was increased in a GTPγS dose-dependent manner (*Figure 6D*). Thus, HMMR interacts with Ran-GTPγS in cell lysates. We predicted that HMMR may affect the localization of active Ran in mitotic cells. To test this, we expressed constitutively active Ran (Ran Q69L, Ran$^{CA}$) (*Kazgan et al., 2010*) in HeLa cells treated with scrambled siRNA or siRNA targeting HMMR. In scrambled siRNA-treated cells, a fraction of Ran$^{CA}$ colocalized with the spindle pole demarked by γ-tubulin (*Figure 6E*), consistent with the identification of Ran in the centrosome proteome (*Andersen et al., 2003*). However, the fraction of Ran$^{CA}$ that colocalized with γ-tubulin, as measured by the ratio of intensities for Ran$^{CA}$ and γ-tubulin, was significantly reduced in HMMR-silenced cells (*Figure 6E–F*). Similarly, inhibition of PLK1 activity, through treatment with a small-molecule inhibitor BI2536, also reduced the fraction of Ran$^{CA}$ that colocalized with γ-tubulin (*Figure 6G–H*). This data shows that reducing PLK1 activity or silencing HMMR reduces the localization of constitutively active Ran at mitotic centrosomes and suggests the phosphorylation of HMMR (pHMMR) by PLK1 may promote pHMMR-RanBP2-Ran-GTP complexes at spindle poles.

## HMMR localizes Ran and positions metaphase spindles in neural cells and tissues

To investigate the HMMR-Ran pathway in neural cells and tissues, we first utilized the neuroblastoma cell line, SHSY5Y, which is known to polarize NuMA during cell division (*Izumi and Kaneko, 2012*). Following the transduction of shRNA targeting HMMR or nonhairpin control shRNA, we confirmed the loss of HMMR immunofluorescence at mitotic spindles (*Figure 7—figure supplement 1A*). In HMMR-silenced cells compared to control-treated cells, we observed a decrease in the immunofluorescence intensity detected for endogenous Ran colocalized with pericentrin, a centrosome marker (*Figure 7—figure supplement 1B–C*). We then investigated mitotic spindle structure and position as well as the localization of Ran in sections derived from WT and *Hmmr$^{tm1a/tm1a}$* E14.5 brains. In these sections stained for the spindle marker beta-tubulin (*Figure 7A*), we measured the position and length of the spindle in dividing neural progenitor cells that lined the ventricles. When compared to spindles within wild-type progenitor cells, spindles in *Hmmr$^{tm1a/tm1a}$* neural progenitors were less centered (a/b measurement) and significantly shorter (c measurement) (*Figure 7B*). Additionally, the density of spindle fibers, as measured by the intensity of beta-tubulin fluorescence, was significantly lower in *Hmmr$^{tm1a/tm1a}$* neural progenitors than those in WT cells (*Figure 7C*). Similar to

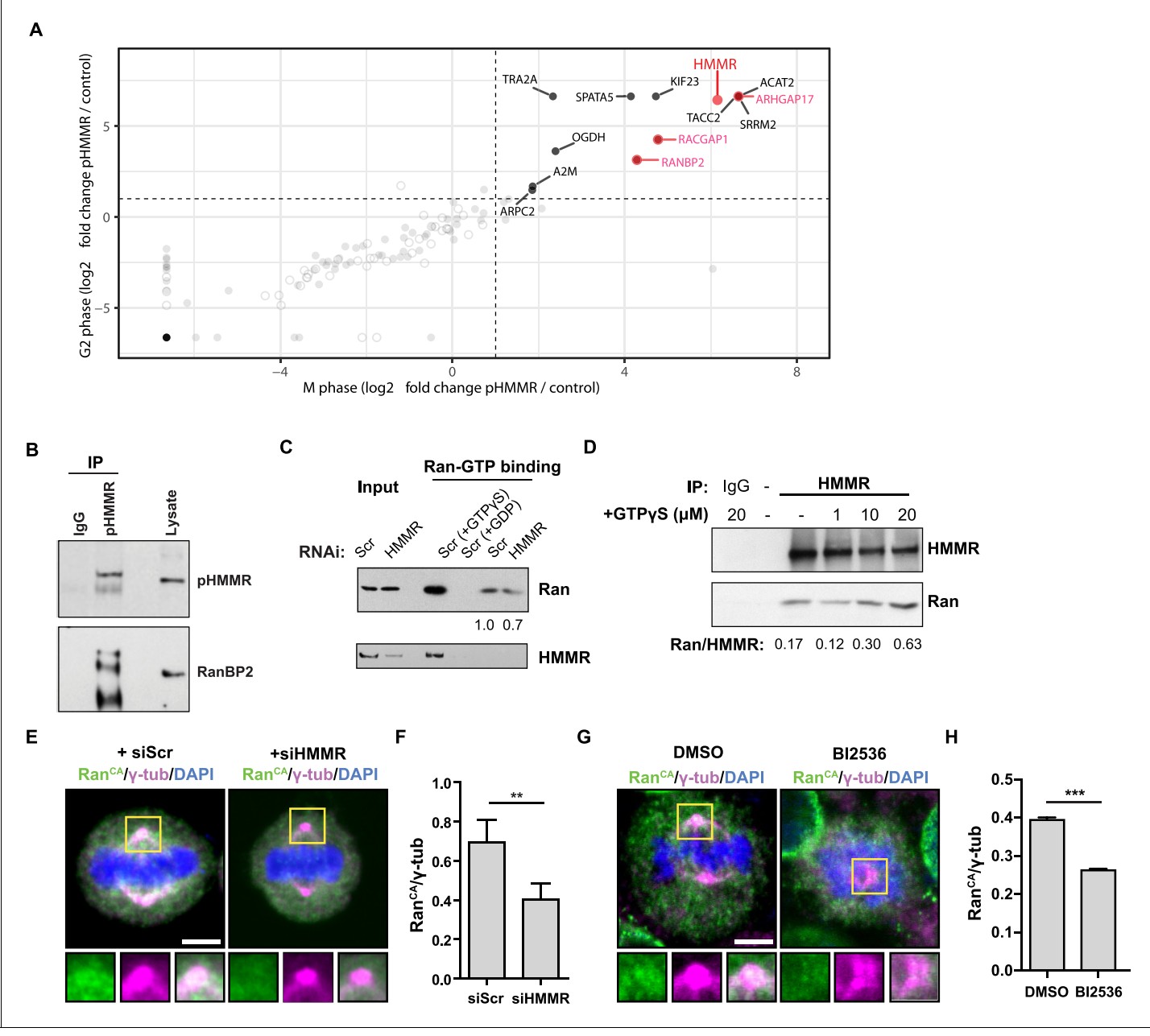

**Figure 6.** Phosphorylated HMMR interacts with RanBP2 and regulates the centrosome localization of Ran. (**A**) Mass spectrometry analysis of proteins co-precipitated from HeLa cell lysates with antibodies targeting HMMR phosphorylated at threonine 703 (pHMMR). Proteins, other than HMMR, co-precipitated with antibodies targeting non-phosphorylated HMMR have been removed. (**B**) Immunoblots of mitotic HeLa extracts subjected to immunoprecipitation with pHMMR antibody or control IgG and blotted with the indicated antibodies. (**C**) Ran activity assay (GTP-bound Ran is immunoprecipitated) in mitotic HeLa cells transfected with scrambled (Scr) or siRNA targeting HMMR (two replicates). (**D**) Immunoblots of mitotic HeLa extracts treated with the indicated concentration of non-hydrolysable GTP (GTPγS) and immunoprecipitated with HMMR antibody or control IgG and blotted with the indicated antibodies (three replicates). (**E**) Localization of constitutively-active Ran (RanQ69L; Ran$^{CA}$) in mitotic HeLa cells transfected with scrambled (Scr) or siRNA targeting HMMR. Yellow boxes indicate magnified region. Scale bar, 5 μm. (**F**) Quantification of the ratio of intensities for Ran$^{CA}$ relative to γ-tubulin in mitotic HeLa cells transfected with scrambled (Scr) or siRNA targeting HMMR. Data are represented as mean ±SD (**p<0.01; 3 replicates of ≥50 cells). (**G**) Localization of constitutively-active Ran (RanQ69L; Ran$^{CA}$) in cells treated with DMSO or BI2536 (Plk1 inhibitor, 20 nM). Yellow boxes indicate magnified region. Scale bar, 5 μm. (**H**) Quantification of Ran$^{CA}$ localization at the centrosome in cells treated with DMSO or BI2536 (20 nM). Data are represented as mean ±SD (***p=0.0006; 2 replicates of >30 cells).

DOI: https://doi.org/10.7554/eLife.28672.016

The following source data and figure supplement are available for figure 6:

**Source data 1.** Data set for mass spectrometry analysis of proteins that interact with pHMMR.

*Figure 6 continued on next page*

*Figure 6 continued*

DOI: https://doi.org/10.7554/eLife.28672.018

**Figure supplement 1.** Total population of proteins complexed with pHMMR.

DOI: https://doi.org/10.7554/eLife.28672.017

our observations in HMMR-silenced HeLa or SH-SY5Y cells, we found a fraction of endogenous Ran, as measured by immunofluorescence, colocalized with a centrosome marker (γ-tubulin, *Figure 7D*); this fraction of Ran, as measured by the ratio of intensities for Ran and γ-tubulin, was significantly reduced in $Hmmr^{tm1a/tm1a}$ relative to WT neural progenitors (*Figure 7E*).

Taken together, our data obtained from studies of cells, tissues and animals that are deficient for HMMR indicate a critical role for the protein in establishing the correct position of the mitotic spindle during cell division. Complete loss of HMMR is sufficient to alter the position and orientation of NP cell division and, in the case of $Hmmr^{tm1a/tm1a}$ mice, disturb brain development and impair the animal's ability to successfully transition to extrauterine life. Our in vitro data support a model where HMMR supports PLK1 activity at the centrosome, which stabilizes astral microtubules and through the phosphorylation of HMMR localizes Ran to mitotic spindle poles. These PLK1-dependent spindle pole positioning processes are critical to reduce cortical localization of NuMA-DHC in off-center spindles and prevent spindle rotation (*Figure 7—figure supplement 2A–B*).

## Ectopic HMMR alters Ran localization, cortical NuMA and spindle positioning

To compare and contrast the mitotic consequences that follow HMMR deletion to those that follow ectopic GFP-HMMR expression, we studied spindle positioning and NuMA localization using HeLa cells with doxycycline-inducible expression of GFP-HMMR (tet-HMMR) (*Figure 8A*), which express GFP-HMMR along the spindle microtubules in mitotic cells and undergo spindle tumbling when grown on L-shaped micropatterned substrates (*He et al., 2017*). In subconfluent cultures of induced tet-HMMR cells, metaphase spindles were more frequently off-center (*Figure 8B*). We next examined whether induced tet-HMMR expression altered the localization of co-expressed $Ran^{WT}$, $Ran^{CA}$, or $Ran^{DN}$ constructs tagged with mCherry. In HeLa cells, these constructs localized as previously reported (*Hutchins et al., 2009*): $Ran^{WT}$ was cytoplasmic, $Ran^{CA}$ localized to the mitotic spindle, and $Ran^{DN}$ localized to the chromosomes (*Figure 8C*). In induced tet-HMMR cells, however, $Ran^{WT}$ localized to the spindle similar to that of $Ran^{CA}$ (*Figure 8C*). Induction of GFP-HMMR expression also caused NuMA to be lost entirely from the cortex (*Figure 8D–E*). We observed a similar effect on cortical NuMA localization in HeLa cells overexpressing $Ran^{CA}$ (*Figure 8F–G*). Taken together, these data indicate that expression of GFP-HMMR induces defects in the spindle positioning pathway that are consistent with an ectopic localization of Ran-GTP on mitotic spindles (*Figure 7—figure supplement 2B*).

## Discussion

HMMR is classified as a non-motor spindle assembly factor (*Manning and Compton, 2008*) and has been shown to be a critical cell division gene product in immortalized cancer cells (*Neumann et al., 2010*). In order to study HMMR functions in vivo, we generated $Hmmr^{tm1a/tm1a}$ mice, which encode a targeting construct following *Hmmr* exon 2. Our approach is in contrast to previous published *Hmmr* mutant mice. An initial murine model, termed $Hmmr^{-/-}$, was generated by targeted disruption at exon 8 (*Tolg et al., 2003*) while an alternative model, termed $Hmmr^{m/m}$, was generated by targeted disruption at exon 10 (*Li et al., 2015*); thus, truncated N-terminal *Hmmr* transcript or protein is expressed in both of these published *Hmmr* models. $Hmmr^{-/-}$ animals exhibited fertility defects (*Tolg et al., 2003*), while $Hmmr^{m/m}$ mice demonstrated deficient spindle orientation during gametogenesis (*Li et al., 2016*; *Li et al., 2015*). However, $Hmmr^{m/m}$ and $Hmmr^{-/-}$ animals are both viable and phenotypically normal. This contrasts with the severely diminished survival we observed in $Hmmr^{tm1a/tm1a}$ neonates and, for the few animals that survived the transition, the reduced body sizes we observed in rare, adult $Hmmr^{tm1a/tm1a}$ mice.

We observed in $Hmmr^{tm1a/tm1a}$ mice previously unseen neural defects such as enlarged ventricles and microcephaly; both of these congenital conditions can arise from defects in spindle orientation

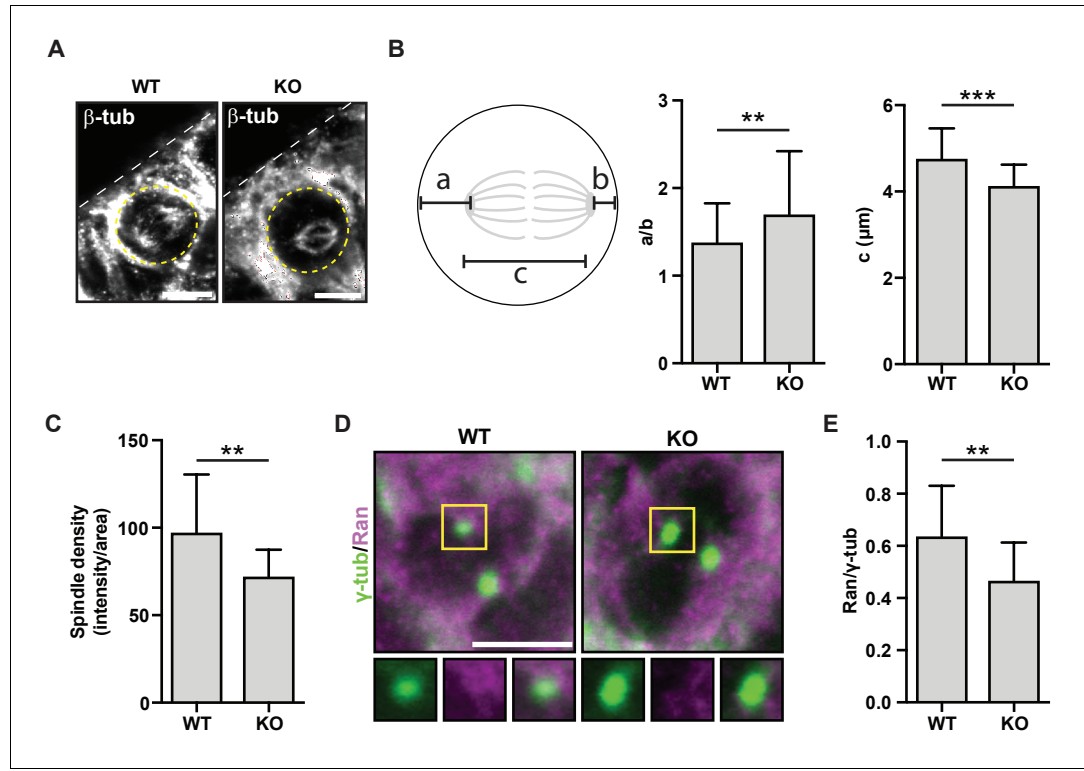

**Figure 7.** HMMR regulates centrosomal Ran and spindle centering in neural tissues. (**A**) Spindle position, density, and length in neuroepithelial progenitor (NP) cells in *Hmmr*^+/+ (WT) or *Hmmr*^tm1a/tm1a (KO) E14.5 mice. Yellow dotted line indicates cell border and white dotted line indicates the ventricle edge. Scale bar, 5 μm. (**B**) Quantification of spindle position and length in WT and KO E14.5 mice. Data are represented as mean ±SD (**p=0.003 (a/b),***p=5.4e-7 (c); n = 57 (WT- a/b), 56 (KO- a/b), 56 (WT- c), 51 (KO- c)). (**C**) Quantification of spindle density in WT and KO E14.5 mice. Data are represented as mean ±SD (**p=0.003; n = 25 (WT), 20 (KO)). (**D**) Localization of Ran in NP cells in WT and KO E14.5 mice. Scale bar, 5 μm. (**E**) Quantification of Ran localization in WT and KO E14.5 brains. Data are represented as mean ±SD (**p=0.002; n = 29 (WT), 49 (KO)).
DOI: https://doi.org/10.7554/eLife.28672.019

The following figure supplements are available for figure 7:

**Figure supplement 1.** Centrosome localized HMMR localizes Ran and positions metaphase spindles in neural cells and rosettes derived from ES cells.
DOI: https://doi.org/10.7554/eLife.28672.020

**Figure supplement 2.** Model of HMMR regulation of spindle positioning via Ran regulation of NuMA-Dynein complexes.
DOI: https://doi.org/10.7554/eLife.28672.021

and planar cell division within the progenitor population (*Lancaster and Knoblich, 2012*). In accordance, we observed a reduced proportion of mitotic Pax6[+] cells that did not properly align their division axis in the subventricular zone (SVZ) and a decrease in the proportion of Tbr2[+] progenitors at E14.5, suggesting an overall decreased neuronal production in these animals. These cellular phenotypes and reduced body size, are similar to those that accompany microcephalic brains observed in Magoh[Mos2/+] mice (*Silver et al., 2010*), which are haploinsufficient for a regulator of the levels of expression of the dynein adaptor protein, Lis1. We also observed enlarged ventricles, which augmented the gross brain size in a proportion of *Hmmr*^tm1a/tm1a mice. The etiology of these phenotypes may relate to impaired cerebrospinal fluid flow, as seen with mutation of dynein components and related to primary cilia dyskinesis (*Ibañez-Tallon et al., 2002*; *Ostrowski et al., 2010*). As HMMR is a non-motor adaptor for dynein, a more detailed analysis of the effect on CSF flow is *Hmmr*^tm1a/tm1a mice is warranted.

A recent study identified misoriented division planes in apical neuroprogenitors and postnatal granule cell precursors, without changes to the cell division rate, during brain development of

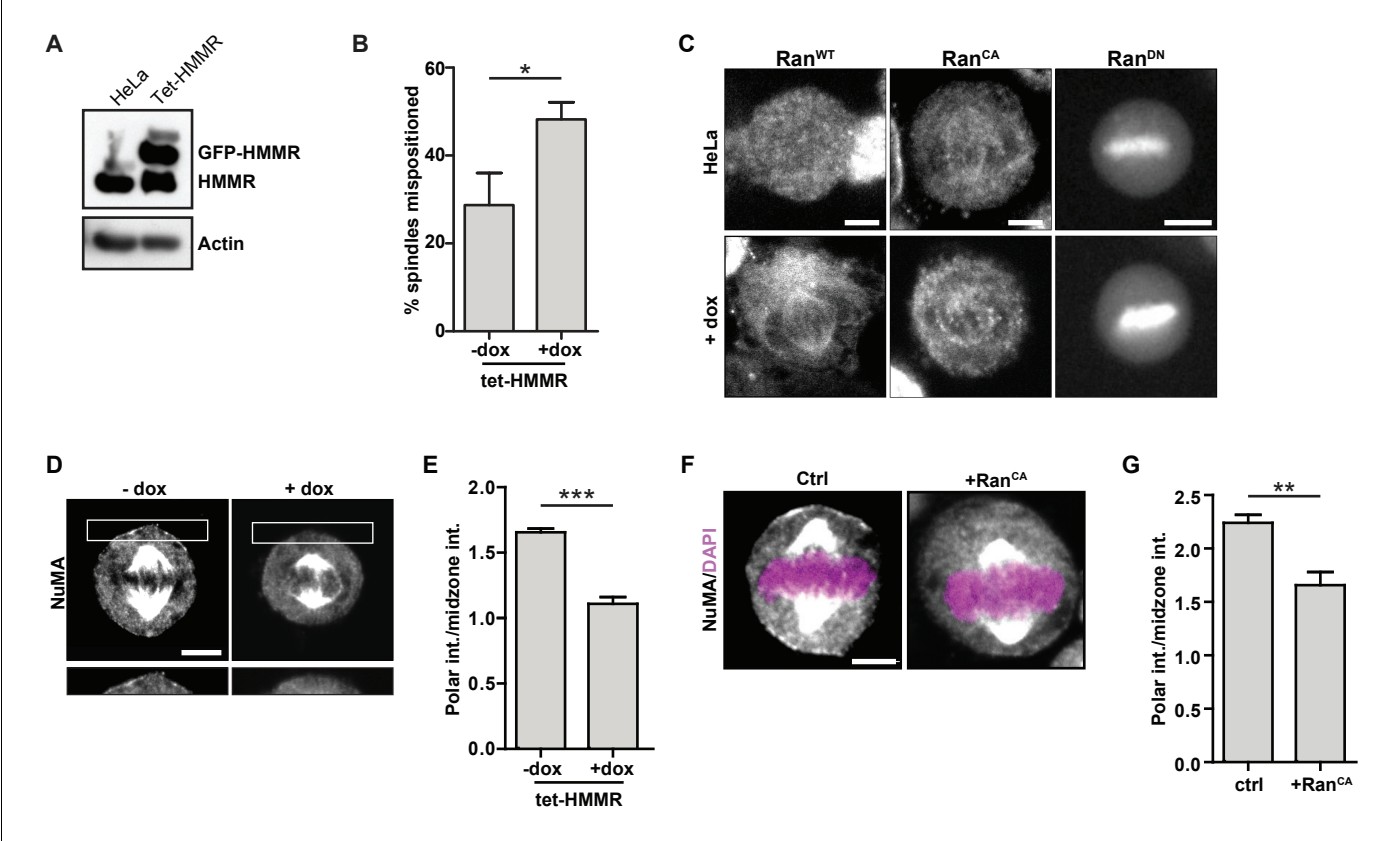

**Figure 8.** HMMR over-expression affects spindle positioning through Ran activation. (**A**) HMMR over-expression in induced tet-HMMR cells. HMMR is over-expressed under the control of a tet-inducible promoter (+dox). (**B**) HMMR over-expression leads to mispositioned mitotic spindles. Data are represented as mean ±SD (p=0.015; 3 replicates of ≥30 cells). (**C**) Localization of RanWT, RanCA, and RanDN constructs in tet-HMMR cells. Scale bars, 5 μm. (**D**) NuMA localization in induced and uninduced tet-HMMR cells. Lower panel shows magnified view of cortex near spindle pole. Scale bars, 5 μm. (**E**) Quantification of NuMA localization in induced and uninduced tet-HMMR cells. Data are represented as mean ±SD (***p=0.0004; 3 replicates of ≥30 cells). (**F**) Localization of NuMA in HeLa cells expressing RanCA constructs. Scale bar, 5 μm. (**G**) Quantification of NuMA localization in HeLa cells expressing RanCA constructs. Data are represented as mean ±SD (p=0.002; 3 replicates of ≥25 cells).

DOI: https://doi.org/10.7554/eLife.28672.022

$Hmmr^{m/m}$ mice expressing truncated N-terminal $Hmmr$ protein (*Li et al., 2017*). Transient megalencephaly was observed in $Hmmr^{m/m}$ brains during the neuronal differentiation period, up to PND7 but normalized by PND14, and attributed to misoriented divisions in apical neuroprogenitors leading to an increased number of Tbr+ intermediate progenitors contributing to cerebral cortex enlargement (*Li et al., 2017*). In $Hmmr^{tm1a/tm1a}$ mice, however, the predominant phenotype observed was microcephaly, which may be explained by a decreased mitotic rate for Pax6$^+$ radial glial cells; consistent with the recent observations in $Hmmr^{m/m}$ brains, we did observe rare $Hmmr^{tm1a/tm1a}$ brains at E14.5 with cortex enlargement.

The different neural phenotypes between $Hmmr^{m/m}$ and $Hmmr^{tm1a/tm1a}$ mice imply the expression of truncated N-terminal $Hmmr$ protein retains critical functions needed for neural development and neonatal survival. In support of this, the microtubule-binding domain of HMMR (*Assmann et al., 1999*; *Maxwell et al., 2003*) has recently been shown to be important for neural development in *Xenopus* (*Prager et al., 2017*). Microinjection of morpholinos targeting $Hmmr$ in *Xenopus laevis* embryos resulted in gross alterations in forebrain morphogenesis and anterior neural tube closure and these defects were rescued by the expression a truncated $HMMR$ construct that preserved the microtubule-binding domain (*Prager et al., 2017*). These defects were attributed to the loss of a non-mitotic role for HMMR in the promotion of neural cell polarization and radial intercalation concomitant with neural tube closure (*Prager et al., 2017*). While we observed no changes in cell morphology in $Hmmr^{tm1a/tm1a}$ MEFs and no defects in the localization of Par3 or ZO-1 in $Hmmr^{tm1a/tm1a}$

NSCs lining the ventricles, future studies should investigate the disruption of Hmmr function in non-mitotic *Hmmr^{tm1a/tm1a}* tissues and cells, such as during directed cell migration needed for cortico-genesis and neural tube closure (*Marín and Rubenstein, 2003* and *Ayala et al., 2007*).

Directed smooth muscle cell migration following balloon catheter injury in rats is reliant upon HMMR (*Silverman-Gavrila et al., 2011*), and HMMR is expressed in cells within the SVZ and rostral migratory stream in the adult mouse brain, where neural progenitor cells persist throughout life (*Lindwall et al., 2013*). While a detailed investigation of non-mitotic processes are needed in *Hmmr^{tm1a/tm1a}* mice, we focused on mitotic processes due to our analysis of *Hmmr* promoter activity and HMMR expression during the directed neural differentiation of mouse embryonic stem cells (ESC). When we derived neural rosette-like structures from a mouse mouse *Hmmr^{BB0166/+}* ESC line (*Figure 7—figure supplement 1D–E*), in which β-geo is inserted following *Hmmr* exon 7 on one allele, or from WT ESCs (*Figure 7—figure supplement 1F*), we found *Hmmr*/HMMR expression was restricted to apical-positioned, cyclin B1-positive rosette-neural stem cells (r-NSCs); moreover, spindle position was disturbed in apical-positioned r-NSCs derived from *Hmmr^{BB0166/+}* mESCs (*Figure 7—figure supplement 1G–H*). Thus, *Hmmr* shows cell cycle-restricted expression in G2 phase and mitotic cells during the process of neural differentiation, and our data are consistent with the defects we observed in *Hmmr^{tm1a/tm1a}* mice being attributed to alterations in neural progenitor expansion and differentiation as determined by changes to the control of spindle orientation and cell division.

*Hmmr^{tm1a/tm1a}* mice exhibited the following phenotypes that are known to arise from defects in spindle orientation and planar cell division: microcephaly (*Gai et al., 2016*), enlarged ventricles (*Dietrich et al., 2009*; *Godin et al., 2010*), and testicular atrophy and fertility defects (*Li et al., 2016*; *Li et al., 2015*). Moreover, in HeLa cells, HMMR is critical to spindle positioning (*Dunsch et al., 2012*). However, other mitotic or post-mitotic processes may also be disturbed in *Hmmr^{tm1a/tm1a}* mice independent of spindle orientation, such as an increased rate of apoptosis, centrosome amplification (*Marthiens et al., 2013*), and decreased division rate (*Caviness et al., 2003*; *Caviness et al., 1995*). In our study, we focused our efforts on addressing many of these additional factors and specifically those related to cell division. TUNEL analysis revealed no increase in apoptosis in *Hmmr^{tm1a/tm1a}* mice at both embryonic and neonate stage. An elevated occurrence of supernumerary centrosomes, as induced for example through overexpression of PLK4, results in embryonic lethality (*Vitre et al., 2015*). Our analysis revealed no evidence of multipolar spindles or centrosome amplification in neural tissues, but we did note an increased frequency of centrosome amplification in *Hmmr^{tm1a/tm1a}* MEFs. In addition, *Hmmr^{tm1a/tm1a}* MEFs require more time to complete the spindle assembly checkpoint when followed through cell division independent of spindle orientation (Unpublished results). However, whether such delays in cell division kinetics are present in *Hmmr^{tm1a/tm1a}* mice, or if an associated reduced rate of cell division contributed to the observed neural defects remain unknown. A loss of control of the cell division axis, however, has been consistently observed in HMMR-silenced HeLa cells (*Dunsch et al., 2012*) and, as described here, in r-NSCs derived from *Hmmr^{BB1066/+}* mESCs, *Hmmr^{tm1a/tm1a}* MEFs, and neural progenitor cells lining the SVZ in brains isolated from *Hmmr^{tm1a/tm1a}* mice.

The intrinsic spindle positioning pathway regulates components of the dynein motor complex on the cortex to generate forces that center spindles. It is well recognized that the PLK1 gradient at both centrosomes and kinetochores is essential in regulating the spindle positioning pathway (*Kiyomitsu and Cheeseman, 2012*; *Tame et al., 2016*). Here, we find HMMR is a critical component of a PLK1-dependent positioning pathway, which contrasts with the findings of a prior report that predicts HMMR-CHICA-DYNLL1 play a PLK1-independent role that regulates the composition of cortical dynein complexes (*Dunsch et al., 2012*). While DYNLL1 and CHICA were identified in our mass spectrometry analysis of pHMMR immunoprecipitates and we found that the incorporation of DYNLL1 into DHC-GFP complexes was dampened in HMMR-silenced cell lysates, our data are consistent with HMMR-DYNLL1 enabling the incorporation of p150-glued into DHC-GFP complexes. This role is similar to that reported for Dyn2 (yeast DYNLL1 ortholog) (*Stuchell-Brereton et al., 2011*) but contrary to the prediction that DYNLL1-bound form of dynein cannot bind to dynein adaptors required for cortical targeting (*Dunsch et al., 2012*).

HMMR acts downstream from Ran-GTP to localize TPX2 (*Groen et al., 2004*; *Joukov et al., 2006*; *Chen et al., 2014*; *Scrofani et al., 2015*), which relies upon a carboxy-terminal bZip motif in HMMR (*Maxwell et al., 2003*) and enables the activation of Aurora A by TPX2 (*Chen et al., 2014*)

(*Scrofani et al., 2015*). As Aurora A can directly phosphorylate PLK1 on Thr210 (*Macůrek et al., 2008*), our observed reduction in pPLK1(T210) in HMMR-silenced cells can be mechanistically explained through reduced Aurora A activity. Consistent with reduced PLK1 activity in HMMR-silenced cells, we find that astral microtubules are reduced and cortical NuMA is retained, which is phenocopied and not additively augmented by PLK1 inhibition. Indeed, the growth rate of EB1-marked microtubule comets was previously shown to be dampened in HMMR-silenced cells although the conclusion of these experiments was that astral microtubule organization was similar to control-treated cells (*Dunsch et al., 2012*). We note that the density of astral microtubules is reduced in HMMR-silenced cells, which provides an alternative mechanistic explanation for the retention of cortical DHC-GFP in HMMR-silenced cells with off-center spindles.

Ran is a component of the centrosome proteome (*Andersen et al., 2003*), although FRET-based assessment of Ran activity has not been seen at the centrosome (*Kaláb et al., 2006*). Here, we find evidence for the location of Ran-GTP at mitotic centrosomes reliant upon PLK1 activity and HMMR. Mass spectrometry analysis discovered RanBP2 as a novel interactor with pHMMR and we show that HMMR precipitates Ran-GTPγS in a dosage-dependent manner and is needed to locate active Ran at mitotic centrosomes. Moreover, in HeLa cells induced to express GFP-HMMR we also observe ectopic localization of Ran to mitotic spindles and the loss of cortical NuMA localization similar to those induced by expression of Ran$^{CA}$. However, HMMR appears dispensable for the Ran-GTP gradient generated at chromosomes, which strips NuMA and DHC-GFP at the midzone cortical region (*Kiyomitsu and Cheeseman, 2012*). Therefore, our data supports a spatially restricted role for pHMMR in the localization of Ran-GTP at centrosomes and spindle fibers downstream of PLK1 activity. However, the relative contribution in the control of spindle positon for pHMMR function with Ran at centrosomes versus the role HMMR plays in promoting pPLK1(T210), potentially through modulation of Aurora A activity, warrants further study.

Taken together, the results of our study identify HMMR as a key component of the PLK1-dependent mitotic spindle positioning pathway that is needed for neural development and neonatal survival. We suggest that the loss of spindle positioning identified here, and by others (*Dunsch et al., 2012*; *Li et al., 2016*; *Li et al., 2015*) (*Li et al., 2017*), in HMMR-silenced and HMMR-deleted or mutated cells and tissues, the occurrence of pleiotropic phenotypes observed in *Hmmr*$^{tm1a/tm1a}$ and *Hmmr*$^{m/m}$ mice, and the requirement of HMMR during meiotic (*Groen et al., 2004*; *Joukov et al., 2006*; *Scrofani et al., 2015*) and mitotic spindle assembly (*Maxwell et al., 2005*; *Maxwell et al., 2003*; *Dunsch et al., 2012*; *Chen et al., 2014*) together indicate that the physiological function for *HMMR* is as a homeostasis, meiosis, and mitosis regulator.

# Materials and methods

### Key resources table

| Reagent type (species) or resource | Designation | Source or reference | Identifiers |
|---|---|---|---|
| genetic reagent (*Mus musculus*) | *Hmmr*$^{tm1a/tm1a}$ | this paper | |
| cell line (*Homo sapiens*) | HeLa | ATCC (Manassas, VA) | CCL-2 |
| cell line (*M. musculus*) | *Hmmr*$^{+/+}$ | this paper | |
| cell line (*M. musculus*) | *Hmmr*$^{tm1a/tm1a}$ | this paper | |
| cell line (*H. sapiens*) | HeLa DHC-GFP | Mitocheck, PMID: 20360068 | |
| cell line (*H. sapiens*) | HeLa tet-HMMR | PMCID: PMC5464802 | |
| cell line (*H. sapiens*) | HeLa eGFP-TUBA, mCherry-histone H2B | Gruneberg lab, PMID: 21187329 | |
| cell line (*H. sapiens*) | SH-SY5Y | Sigma (St. Louis, MO) | 94030304-1VL |
| cell line (*M. musculus*) | ES cells *Hmmr*$^{+/+}$ | Mutant Mouse Resource Regional Cente (MMRC)r | *E14*TG2a |
| cell line (*M. musculus*) | ES cells *Hmmr*$^{BB0166/+}$ | MMRC | 026467-UCD |
| cell line (*M. musculus*) | ES cells *Hmmr*$^{tm1a(EUCOMM)Hmgu}$ | European Conditional Mouse Mutagenesis Program | |
| transfected construct | HMMR siRNA 5'A | Dharmacon (Lafayette, CO), PMID 24875404 | |
| transfected construct | HMMR siRNA 5'B | Dharmacon, PMID 24875404 | |

*Continued on next page*

*Continued*

| Reagent type (species) or resource | Designation | Source or reference | Identifiers |
|---|---|---|---|
| transfected construct | HMMR siRNA 3' | Dharmacon, PMID 24875404 | |
| antibody | Actin | Sigma | a5060 |
| antibody | alpha-tubulin | Abcam (United Kingdom) | ab4074 |
| antibody | beta-tubulin | Sigma | T5293 |
| antibody | beta tubulin-647 | Cell Signaling Technology (Danvers, MA) | 3624 |
| antibody | mCherry | Abcam | ab167453 |
| antibody | DIC | Millipore (Burlington, MA) | mab1618 |
| antibody | DynLL1 | Abcam | ab51603 |
| antibody | EB1 | Abcam | ab53358 |
| antibody | gamma-tubulin | Sigma | t6557 |
| antibody | GFP | Abcam | ab1218 |
| antibody | HMMR | Abcam | ab124729 |
| antibody | pHMMR | PMID:22110403 | |
| antibody | NuMA | Abcam | ab36999 |
| antibody | NuMA | Cell Signaling Technology | 3888 |
| antibody | p150 glued | Abcam | ab151184 |
| antibody | Par3 | Millipore | 1330 |
| antibody | Pax6 | Covance (Princeton, NJ) | prb278p10 |
| antibody | Pericentrin | Covance | prb432c |
| antibody | pPLK1 | Abcam | ab189139 |
| antibody | PLZF | EMD Chemical | op128 |
| antibody | Ran | Cell Biolabs Inc (San Diego, CA) | in STA-409 kit |
| antibody | Ran | Abcam | ab53775 |
| antibody | RanBP2 | Abcam | ab197044 |
| antibody | Tbr2 | Abcam | ab183991 |
| antibody | TPX2 | Novus Biologicals (Littleton, CO) | mb500179 |
| antibody | ZO-1 | Invitrogen (Carlsbad, CA) | 339100 |
| antibody | ZO-1 | Invitrogen | 402200 |
| recombinant DNA reagent | pmCherry-C1-RanQ69L | Addgene (Cambridge, MA), PMID: 20685962 | Addgene #30309 |
| recombinant DNA reagent | pcDNA-RanWT-mRFP1-polyA | Addgene, PMID: 24908396 | Addgene #59750 |
| recombinant DNA reagent | mCherry-Ran$^{DN}$ | PMID: 23870127 | |
| recombinant DNA reagent | HMMR-FL | PMID: 12808028 | |
| recombinant DNA reagent | HMMR 1–623 | PMID: 12808028 | |
| recombinant DNA reagent | HMMR 1–679 | PMID: 12808028 | |
| commercial assay or kit | Ran Activation Assay kit | Cell Biolabs, Inc | STA-409 |
| chemical compound, drug | BI2536 | Selleck Chemicals (Houston, TX) | S1109 |

## Generation of *Hmmr$^{tm1a/tm1a}$* mice

*Hmmr$^{tm1a/+}$* ES cell strains with the L1L2_Bact_P cassette inserted after *Hmmr* exon 2 were purchased from The European Conditional Mouse Mutagenesis Program (HEPD0778_4_B11; EUCOMM). *Hmmr$^{tm1a/+}$* ES cells were introduced into blastocyst stage embryos by microinjection and resulting male chimera mice were bred to C57BL/6J female to obtain *Hmmr$^{tm1a/+}$* mice. *Hmmr$^{tm1a/+}$* mice were then intercrossed to generate *Hmmr$^{tm1a/+}$* mice on a C57BL/6J background. All mice were maintained in the pathogen-free Centre for Molecular Medicine and Therapeutics animal facility on a 6 am-to-8 pm light cycle, 20 ± 2°C, with 50 ± 5% relative humidity, and had food

and water *ad libitum*. All procedures involving animals were in accordance with the Canadian Council on Animal Care (CCAC) and UBC Animal Care Committee (ACC) (Protocol no. A13-0168).

## Genotyping

Tail clips from embryos and ear notches from weaned animals were lysed in GB buffer (100 mM Tris, pH8.8, 100 mM $(NH_4)_2SO_4$, 100 mM $MgCl_2$, 1% β-mercaptoethanol, 0.5% triton X-100 and 1.6 mg/ml protease K) at 50℃ for 3–5 hr. Protease K was inactivated at 95℃ for 10 min. PCR was performed using AccuStart II PCR mix (Quanta Biosciences, Beverly, MA). Primers used for PCR1 were: *Hmmr*-for, 5'-AGATACAACCTTGCTTGCTTCGGC-3', loxR, 5'-TGAACTGATGGCGAGCTCAGACC-3' (mutant 507 bp); Primers used for PCR2 were: *Hmmr*-5'-arm, 5'-CAGGCCTTAGAAGCTGACA TGAGC-3', *Hmmr*-3'-arm, 5'-TCCAAACTTCTCACTGCAGACAGC-3', LAR3, 5'-CAACGGGTTCTTC TGTTAGTCC-3' (WT 515 bp, mt 339 bp).

## Immunoblotting

Mouse tissues were harvested, snap frozen, ground into powder, lysed in RIPA buffer supplemented with protease inhibitor (Roche, Switzerland) and sonicated. Lysates were clarified by centrifugation at 16,000xg for 20 min at 4℃, and concentration was determined by BCA protein assay kit (Thermo Scientific, Waltham, MA). Lysates were mixed with SDS sample buffer, separated by SDS-PAGE, and blotted with the indicated antibodies: Actin (rabbit (rb), Sigma, 1:2500) and HMMR (rb, Abcam, 1:500).

## Cell culture

$Hmmr^{BB0166/+}$ mouse ES cells (BB0166; MMRRC: 026467-UCD) and the parental control mouse ES cells (*E14*TG2a; MMRRC: 015890-UCD) were purchased from a Mutant Mouse Regional Resource Center (University of California, Davis). ES cells were maintained on mitomycin C-treated MEFs prepared as previously described (*Conner, 2001*). ES cells were cultured and neural induction was initiated as previously described (*Barberi et al., 2003*).

To generate $Hmmr^{tm1a/tm1a}$ MEFs, heterozygous mice were interbred as previously described (*Johnson et al., 1995*) with the following exceptions: embryos were collected at day 13.5 (day 1) and homogenized using an 18-gauge needle and a 10cc syringe, cultures were incubated for two days, trypsinized on day 3, and frozen once confluent.

HeLa cells (ATCC: CCL-2) were maintained as previously described (*Chen et al., 2014*). HeLa cells expressing mouse DHC-GFP were obtained from Mitocheck and maintained as described (*Hutchins et al., 2010*). HeLa cells expressing mCherry-Histone H2B, eGFP-TUBA were obtained from the Gruneberg lab (University of Liverpool) (*Zeng et al., 2010*) and maintained in media with 0.3 µg/ml puromycin (Invitrogen) and 0.5 µg/ml blasticidin S (Invitrogen). Live imaging was performed using Leibovitz's L-15 media supplemented with 10% FBS. BI2536 (Selleck Chemicals) treatment was performed for 2 hr at a concentration of 20 nM.

HeLa cells with tet-on inducible expression of enhanced GFP fused in frame with full-length HMMR (GFP-HMMR) were obtained from Dr. LM Pilarski (University of Alberta) and produced as described (*He et al., 2017*). For experiments, expression of HMMR was induced using 2 µg/ml of doxycycline (Clontech, Mountain View, CA) for 10 hr.

SH-SY5Y (Sigma 94030304; ECACC validated prior to purchase) were maintained as previously described (*Izumi and Kaneko, 2012*).

## Immunostaining

Neural rosettes and MEFs were grown on coverslips coated with 0.1% gelatin and fixed with 4% paraformaldehyde (PFA) for 15 min. For pericentrin, cells were fixed with 4% PFA followed by MeOH for 15 min at −20℃. HeLa cells were fixed in ice-cold MeOH for 15 min at −20 ℃ Cells were blocked with 0.3% triton X-100, 10% donkey serum, 0.1% BSA in PBS (rosettes) or 0.3% triton X-100, 1.0% BSA, in PBS. Cells were incubated with primary antibodies for 2 hr at RT or overnight at 4℃ and secondary antibodies for 1 hr at RT.

Mouse tissues were fixed in 4% PFA overnight and stored in 70% EtOH. Tissues were paraffinized, embedded, and sectioned at 5 µm intervals. Deparaffinization and antigen-retrieval were performed and sections were processed for immunostaining as previously described (*Li et al., 2015*). Primary

Antibodies Used: α-tubulin (rabbit (rb), Abcam, 1:1000); β-tubulin (mouse (ms), Sigma); β-tubulin-647 (rb, Cell Signaling, 1:500); EB1 (rat, Abcam, 1:500.); γ-tubulin (ms, Sigma, 1:2000); GFP (ms, Abcam, 1:500); HMMR (rb, Abcam, 1:500; [*Li et al., 2015*])); NuMA (rb, Abcam, 1:500–1000); Par3 (rb, Millipore, 1:100); Pax6 (rb, Covance, 1:300); Pericentrin (rb, Covance, 1:500); pPLK1(Thr210) (rb, Abcam,1:200); PLZF (ms, EMD chemical, 1:100); Ran (ms, Cell Biolabs, 1:500), Ran (rb, Abcam, 1:50); Tbr2 (rb, Abcam, 1:100); TPX2 (rb, Novus, 1:500); ZO-1 (rb, Invitrogen, 1:1000); ZO-1 (ms, Invitrogen, 1:100). Secondary Antibodies Used: AlexaFluor 488, AlexaFluor 549, and AlexaFluor 647 (Invitrogen).

For TUNEL, samples were stained with the In Situ cell death detection kit, Fluorescein (Roche) following the manufacturer's instructions.

Coverslips were mounted with Prolong Gold antifade reagent with DAPI (Invitrogen) and images were acquired with confocal microscopy (FluoView Fv10i, Olympus (Japan) or Axiovert 200, Zeiss (Germany)). Image analysis was performed using ImageJ.

## Live imaging

For DHC-GFP HeLa, imaging was performed using a Perkin Elmer Ultraview VOX spinning disk confocal microscope using a Leica DMI6000 inverted microscope equipped with a Hamamatsu 9100–02 camera. Images were taken at two intervals. MEFs, Tet-HeLa cells, and eGFP-TUBA HeLa cells were imaged using an ImageXpress Micro High Content Screening System (Molecular Devices, Inc., Sunnyvale, CA) for up to 24 hr at 15 min intervals. Prior to imaging cells were stained with Hoechst. Image analysis was performed using ImageJ.

## Virus packaging and transduction

*HMMR* constructs were delivered and expressed using the Gateway system (Invitrogen). Briefly, HEK293FT cells (Invitrogen R7007) were transfected using Lipofectamine 2000 (Invitrogen) with vectors containing, $HMMR^{FL}$, $HMMR^{1-623}$, or $HMMR^{1-679}$, for 72 hr. Supernatant was collected and concentrated using Lenti-X concentrator (Clontech). Virus was added to MEFs with polybrene (8 µg/ml). After 24 hr, the media was replaced and cells grown for 24 hr prior to imaging.

## Transfection

On-target plus siRNA (Dharmacon) and scrambled siRNA as previously described (*Chen et al., 2014*). pmCherry-C1-RanQ69L was a gift from Jay Brenman (Addgene plasmid # 30309) (*Kazgan et al., 2010*). pcDNA-RanWT-mRFP1-polyA was a gift from Yi Zhang (Addgene plasmid # 59750) (*Inoue and Zhang, 2014*). The mCherry-Ran$^{DN}$ construct was graciously provided by Dr. Iain Cheeseman (*Kiyomitsu and Cheeseman, 2013*). Transfection of DNA and siRNA used JetPrime (Polyplus Transfection, France) following the manufacturer's protocols. Cells were harvested 96 hr post-transfection of siRNA.

## Ran activity assay

HeLa cells were treated with siRNA and synced with a double thymidine block and treated with MG132 for 2 hr starting 8 hr post-release. Ran activity assays were performed using the Ran Activation Assay kit (Cell Biolabs, Inc) as per manufacturer's protocols.

## Co-immunoprecipitation

DHC-GFP HeLa cells treated with siRNA and synced with a double thymidine block were homogenized with lysis buffer (*Dunsch et al., 2012*) with a phosphatase inhibitor (Phosphostop (Roche)) and Protease Inhibitor cocktail (Roche). HeLa cells were treated with indicated plasmids and synced with a nocodazole block and then released into mg-132. Cells were lysed with lysis buffer (25 mM HEPES, pH 7.5, 150 mM NaCl, 1% NP-40, 10 mM $MgCl_2$, 1 mM EDTA, 2% glycerol). GTPγS was loaded prior to immunoprecepitation by adding EDTA (10 µM) and GTPγS at the indicated concentration for 30 min at 30°C. The reaction was stopped with $MgCl_2$ (65 µM). The supernatants were used for immunoprecipitation with IgG (ms and rb, Sigma), GFP (ms, Abcam), HMMR (rb, Abcam), or pHMMR (*Maxwell et al., 2011*), overnight at 4°C, followed by incubation with protein A/G beads overnight at 4°C (Santa Cruz Biotechnology). Protein A/G beads were washed with lysis buffer three times. Bound proteins were separated by SDS-PAGE and analysed by western blotting. Actin (rb, Sigma,

1:2500); mCherry (rb, Abcam, 1:1000); HMMR (rb, Abcam, 1:500); GFP (ms, Abcam, 1:1000); DIC (ms, Millipore, 1:500); DynLL1 (rb Abcam, 1:1000); p150$^{Glued}$ (ms, Abcam, 1:500); Ran (ms, Cell Biolabs Inc, 1:1000); and RanBP2 (Abcam, 1:5000).

## Mass spectrometry of protein complexes

Cells were lysed at $0.5–1.0 \times 10^7$ cells/ml in immunoprecipitation buffer (50 mM Tris-HCl, pH 7.4, 150 mM NaCl, 1 mM EDTA, 0.5% NP-40) supplemented with protease and phosphatase inhibitors (Roche). Cell lysates were clarified by centrifugation at 16,000 X $g$ for 10 min at 4°C and protein concentration was determined using the BCA protein assay kit (Thermo Fisher). For immunoprecipitation, lysates were precleared with protein G or A/G PLUS-Agarose beads (Santa Cruz). Protein complexes were isolated by incubation with the indicated antibodies at 4°C on rotation, and then with protein G or A/G PLUS-Agarose beads for 6 hr at 4°C on rotation. Isolated complexes were washed four times with lysis buffer.

Following IP, protein samples on beads were eluted twice with 50 µL of 100 mM citric acid, pH 2.6 at 50°C for 10 min shaking at 1300 rpm, followed by centrifugation, collection of the supernatant and neutralization with 125 µL of 1 M HEPES, pH 8.5. Proteins were reduced by adding 5 µL of 200 mM DTT and incubating at 37°C for 60 min, followed by alkylation by adding 10 µL of 400 mM IAA and incubation at room temperature for 60 min in the dark. The reaction was quenched by adding 10 µL of 200 mM DTT. Proteins were digested with Trypsin/Lys-C mix (Promega, Madison, WI) at an enzyme:protein ratio of 1:100 at 37°C for 16 hr. For stable isotope labeling by reductive dimethylation formaldehyde and heavy formaldehyde ($C^{13}D_2O$) was added to 40 mM final concentration to IgG control and HMMR IP samples, respectively. Sodium cyanoborohydride was added to a final concentration of 20 mM immediately after to both samples and incubated at 21°C for 60 min. Both conditions were combined, acidified to pH 2.5 with TFA, and the peptides purified with C18-STAGE tips as described (*Rappsilber et al., 2003*).

Liquid chromatography tandem mass spectrometry analysis was performed with the Easy nLC ultra-high-pressure LC system (Thermo Fisher Scientific) couple to a Q Exactive HF mass spectrometer with an EASY-Spray source. An EASY-Spray C18 column (Thermo-Fisher, 50 cm long, 75 µm inner diameter) heated to a temperature of 50°C was used for separation. Dried Stage-tip eluates were resuspended in 10 µL buffer A (0.1% FA) and 2 µL was used for injection. The peptides were loaded at a back pressure of 550 bar and separated with a gradient of 3–25% buffer B (0.1% FA in 80% ACN) over 105 min followed by 25–40% buffer B over 20 min at a flow rate of 300 nL/min. The chromatography method ends with a ramp from 40 to 100% buffer B over 3 min then a hold at 100% buffer B for 12 min. A column equilibration step using 11 µL buffer A was included prior to the next sample loading step.

MS data were acquired using a data-dependent top 12 method with a dynamic exclusion of 20 s. MS1 was performed at a resolution of 60,000 at m/z 200 with an AGC target of 3E6 and a maximum ion injection time of 75 ms over the m/z range 400 to 1800. HCD fragmentation of peptides was performed with an isolation range of 1.4 m/z and normalized collision energy set to 28. A resolution of 15,000 at m/z 200, an AGC target of 5E4 and a maximum ion injection time of 50 ms was set for fragment spectra acquisition.

Acquired spectra from two separate experiments and multiple injections were searched using Proteome Discoverer 2.1 (Thermo Fisher Scientific). Database search was performed against the *Homo sapiens* reference proteome including isoforms downloaded from UniProt in June 2016. Main search parameters: enzyme: Trypsin (full); missed cleavages: 2; precursor mass tolerance: 10 ppm; fragment mass tolerance: 0.02 Da; static modifications: +57.021 Da on C; variable modifications: +15.995 Da on M, +28.031 or +34.063 on K and peptide N terminus, +42.011 Da on protein N terminus. Identifications were filtered for 1% FDR at the peptide and protein level. Differential abundance of proteins between pHMMR and control IP was calculated based on the area of heavy and light dimethyl precursor peaks. Common contaminants and decoy identifications were filtered out. To identify proteins commonly identified in affinity purification experiments the identified proteins were searched against the CRAPOME database (www.crapome.org, version 1.1, H. sapiens). Proteins that were found in less than 30% of reported experiments were classified as 'rare', the reminder as 'common'. Proteins not matched in the CRAPOME database were classified as 'unknown'. Full Proteome Discoverer results are available in *Figure 6—source data 1* and raw data is available through the PRIDE Archive.

## Quantification of mitotic spindle orientation

Neural rosettes were stained with TPX2, ZO-1, and TUBB and images were acquired using confocal microscopy. In rosettes, spindle orientation was measured as the angle of the cleavage plane (anaphase or telophase cells) relative to the apical surface. For analysis of the division orientation of apical NP cells, E14.5 brain sections were stained for pH3 and γ-Tubulin. The long spindle axis of anaphase cells, defined by a line bisecting the two centrosomes, was used to indicate the cell division plane. For each progenitor, the angle between the long spindle axis and the apical surface (defined by a line along the centrosomes of apically localized interphase cells) was determined. In cultured MEFs, spindle orientation was measured between the long axis of G2 phase cells (15 mins prior to mitosis) and the angle of the mitotic spindle (anaphase cells).

## Statistical methods

All replicates were biological replicates. Statistical analysis was performed using GraphPad Prism v5.01 for Windows (Graphpad Software, La Jolla, CA). Pairwise comparisons were made using two-tailed, paired or unpaired student's t-test. Comparisons of multiple groups were made using one-way analysis of variance (ANOVA) with a Bonferroni post-test.

## Acknowledgements

We thank Dr. Iain Cheeseman (MIT) for reagents, the UBC Bioimaging Facility, The Centre for Phenogenomics (Toronto, Canada), and the FLI Histology and Imaging Core Facilities. This work was funded by: Michael Cuccione Foundation (studentship, fellowship, or salary awards to HC, MC, PFL, or CAM), Canadian Breast Cancer Foundation (PhD fellowship to TC), BC Children's Hospital Research Institute (studentship to ZH, and salary awards to PFL and CAM), Canada Research Chairs (salary award to PFL), Michael Smith Foundation for Health Research (salary award to PFL), and Canadian Institutes of Health Research (OBC 134036 to CAM; salary award to CAM).

## Additional information

### Funding

| Funder | Grant reference number | Author |
|---|---|---|
| Michael Cuccione Foundation | | Marisa Connell<br>Helen Chen<br>Christopher A Maxwell<br>Philipp F Lange |
| Canadian Breast Cancer Foundation | | Tony Chu |
| BC Children's Hospital Research Institute | | Zhengcheng He<br>Philipp F Lange<br>Christopher A Maxwell |
| Canada Research Chairs | | Philipp F Lange |
| Michael Smith Foundation for Health Research | | Philipp F Lange |
| Canadian Institutes of Health Research | OBC 134038 | Christopher A Maxwell |

The funders had no role in study design, data collection and interpretation, or the decision to submit the work for publication.

### Author contributions

Marisa Connell, Conceptualization, Formal analysis, Investigation, Writing—original draft, Writing—review and editing; Helen Chen, Conceptualization, Formal analysis, Investigation, Writing—review and editing; Jihong Jiang, Formal analysis, Investigation, Writing—original draft, Writing—review and editing; Chia-Wei Kuan, Abbas Fotovati, Tony LH Chu, Zhengcheng He, Tess C Lengyell, Huaibiao Li, Torsten Kroll, Amanda M Li, Daniel Goldowitz, Lucien Frappart, Millan S Patel, Investigation,

Writing—review and editing; Aspasia Ploubidou, Philipp F Lange, Supervision, Investigation, Writing—review and editing; Linda M Pilarski, Resources, Writing—review and editing; Elizabeth M Simpson, Resources, Investigation, Writing—review and editing; Douglas W Allan, Resources, Supervision, Writing—review and editing; Christopher A Maxwell, Conceptualization, Resources, Supervision, Funding acquisition, Writing—original draft, Writing—review and editing

### Author ORCIDs
Marisa Connell, http://orcid.org/0000-0002-7594-2137
Huaibiao Li, http://orcid.org/0000-0003-4086-3321
Christopher A Maxwell, https://orcid.org/0000-0002-0860-4031

### Ethics
Animal experimentation: All procedures involving animals were in accordance with the Canadian Council on Animal Care (CCAC) and UBC Animal Care Committee (ACC) (Protocol no. A13-0168).

### Decision letter and Author response
Decision letter https://doi.org/10.7554/eLife.28672.026
Author response https://doi.org/10.7554/eLife.28672.027

## Additional files

### Supplementary files
• Transparent reporting form
DOI: https://doi.org/10.7554/eLife.28672.023

### Major datasets
The following dataset was generated:

| Author(s) | Year | Dataset title | Dataset URL | Database, license, and accessibility information |
|---|---|---|---|---|
| Connell M, Chen H, Jiang J, Kuan C-W, Lange PF, Maxwell CA | 2017 | phospho HMMR Co-Immunoprecipitation | https://www.ebi.ac.uk/pride/archive/projects/PXD007897 | Publicly available at the EBI European Nucleotide Archive (accession no: PXD007897) |

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
