## [Decision Letter]

Thank you for submitting your article "*HMMR* localizes active Ran at the centrosome to orient cell division and support neural development" for consideration by *eLife*. Your article has been reviewed by three peer reviewers, and the evaluation has been overseen by a Reviewing Editor and Anna Akhmanova as the Senior Editor. The following individual involved in review of your submission has agreed to reveal his identity: Iain M Cheeseman (Reviewer #1).

The reviewers have discussed the reviews with one another and the Reviewing Editor has drafted this decision to help you prepare a revised submission.

Summary:

In this paper, the authors investigate the role of the HMMR protein in mitotic spindle positioning and orientation. The authors use multiple systems (mutant mouse, ES derived rosettes, fibroblasts derived from mutant mice, HeLa cells, neuroblastoma cells and *Drosophila*) to demonstrate the function of HMMR in spindle orientation. Although the role of HMMR in spindle orientation is not new, the authors have identified mechanism by which HMMR controls spindle positioning in cells. They find that reducing or eliminating HMMR perturbs spindle positioning and alters the cortical localization of both dynein and NuMA. They provide evidence that this occurs by HMMR associating with Ran to alter this signaling pathway.

Essential revisions:

The reviewers are in agreement that this paper is interesting, but feel that there are several things that are necessary to improve the paper to the point where it would be acceptable for publication in *eLife*. The primary suggestions relate to strengthening the mechanistic aspects of the work.

1) Your paper suggests an important role for spindle pole-localized HMMR in signaling to the cell cortex to promote proper spindle positioning and regulating DHC and NuMA localization. Prior work has suggested that Plk1 also acts as a spindle-localized signal to control spindle position. Does this suggest that HMMR impacts Plk1 activity as well? It would be very useful to consider the relationship between these two pathways and compare the effects from individually or simultaneously targeting these pathways:

• It is important test whether HMMR affects Plk1 signaling. The reviewers recommend that you combine Plk1 inhibition or cortically tethering with HMMR perturbations for the experiments in Figure 5 and 6.

• The authors show that HMMR centrosome localization domain is important for its function in spindle orientation. This is a nice part of the study, and the authors use many different techniques to decipher this mechanism. However, as they admit in the Discussion, the relative importance of HMMR-Ran-centrosomal pathway with respect to others involving Plk1 and DYNLL1 is not determined; making less clear what is the relevance of these results. The authors may want to dissect whether Ran centrosome localization alone is important for spindle orientation by modulating centrosomal Ran levels independently of HMMR and by altering DYNLL1 and PlK1 expression and activity in the HMMR overexpression experiments.

2) It would be helpful to more carefully evaluate the role of Ran in this process using Ran mutants (constitutively active or dominant negative). For example, does a Ran constitutively active mutant phenocopy HMMR overexpression? Alternatively, does Ran inhibition suppress the effects of HMMR overexpression? For the work on Ran, it would provide a useful addition if you could provide data to clarify the interaction and nature of the relationship between HMMR and Ran (direct interaction, through RanBP1, does this affect Ran activity, etc.):

• siHMMR cells still deplete NuMA from the lateral cortex, suggesting this chromosomal Ran effect is functional. The authors might consider including additional siRNA conditions, such as against Rcc1 alone or in combination with HMMR, to compare HMMR to loss of activated Ran. A discussion of how HMMR discriminates potential subcellular pools of Ran as well as how it may or may not interfere with the Plk1-Dynein regulatory loop would be beneficial.

• To further establish the link between the HMMR-Ran centrosomal pathway described in vitro and in vivo observations (and to confirm that the HMMR-centrosomal Ran pathway controls spindle orientation to regulate neurogenesis), the authors should show localization of Ran to centrosomes in vivo (the fact that they study this in neuroblastoma cells is not sufficient).

• It is also important to consider alternative explanations for the observed phenotypes beyond altered Ran signaling.

- HMMR loss is shown to amplify centrosome number in MEFs (although a discussion of how this might occur is not presented). Have the authors considered that this centrosome amplification may contribute to spindle orientation defects (e.g. see Kulukian et al. PNAS 2015) independent of any HMMR-Ran connection?

- To what extent is the orientation pathway described in vitro responsible for orientation defects in vivo? Polarity defects could affect spindle orientation upstream of the force localizing machinery (which is different from the mechanism the authors propose). Spindle density, mitotic defects and centrosome amplification phenotypes should be also considered when analyzing /stating the mechanism controlling spindle orientation. Are Astral MT affected? Is mitosis length increased? In spindle pictures in vivo and in ES derived rosettes, spindle density and size seem affected. The authors should characterize these features (in vivo and /or in cells). Generally it would be useful to analyze polarity markers, centrosome amplification, spindle morphology in vivo (or at least in rosettes).

3) The reviewers also recommend that you trim aspects of your analysis to focus on the novel and substantive new insights. All three reviewers expressed confusion for the diverse data and systems, some of which was not directly relevant to the overall conclusions. The authors could do a better job of integrating this data and experimental systems. The reviewers recommend removing some of the data from the work in *Drosophila* cells and other experiments that are not directly related to the main points of the paper, although we leave this to your discretion.

• In its current version, it seems that this paper is constructed so that it can accommodate a large amount of data obtained in different systems. The authors try to squeeze together a neurodevelopmental biology study and a cell biology study, and the resulting message is confused and confusing. The paper is difficult to read, and it needs a major overhaul.

• The inclusion of the *Drosophila* experiments seems unnecessary as there is no complementation of the Miranda phenotype (perhaps not surprising given the evolutionary difference), and they only use this to see cortical blebbing. If the authors want to include this data, they could alter this section to minimize the analysis of the complementation (perhaps a single sentence).

4) The in vivo analysis with the mouse model would benefit from some additional consideration. For the revised paper, you should use caution in the interpretation of the mouse knockout phenotypes and avoiding redundancy with other recently published work to highlight your novel discoveries:

• Previous work by Li et al. also generated a HMMR mouse mutant. There are a number of similarities, but also differences between these two papers. In the previous case, the authors only eliminated the C-terminal region of the protein. Here, the authors argue that the lethality of their mutant (but not the previous mouse model) is related to functions of the N-terminus. However, the prior paper also analyzed spindle orientation with apparently similar observations. It would be helpful to further clarify the differences and similarities between the two papers.

• Although the systems are used somewhat interchangeably, in some cases they are not used to their full potential. The power of this paper to generate a potent knockout for HMMR in mice. However, they do not use cells derived from these mice for most of the subsequent cell biological experiments. Instead, they deplete the protein by RNAi in HeLa cells, which may be both less efficient and also prone to off target defects (no rescue is shown).

• The authors state that HMMR effect is "pleiotropic" and are careful not to conclude that the observed developmental defects derive strictly from spindle orientation defects. However a large part of the Introduction is based on the idea that spindle orientation dictates cell fates in the developing CNS, and somehow directs the reader towards this interpretation. This is actually a very dogmatic view: although many published studies claim a causal relationship, they only show a correlation between strong neurogenesis defects (that may be caused by defective migration, cell death, or many other factors) and often minor defects in spindle orientation. As the authors later show in different HMMR mutant contexts, there are actually many defects:

- increased cell death,

- probable aneuploidy,

- centrosome mispositioning (described in rosettes),

- centrosome overduplications (described in MEFs)

- mitotic cell death (in Figure 5)which are as many possible causes of microcephaly. If all these defects occur also in vivo (which they do not document), then one cannot conclude that neurogenic and brain formation defects are specifically generated by spindle misorientation itself. Indeed, centrosome amplification has been linked to microcephaly independently of spindle orientation (Marthiens et al., 2013). Centrosome positioning defects in rosettes suggest that polarity may be affected, and indeed the authors cite defects in a-b in mammary epithelia. Polarity defects can affect the architecture of the neuroepithelium, affecting indirectly neuroepithelium structure and neurogenesis, and it would be necessary to investigate a-b cell polarity in mutant brains.

[Editors' note: further revisions were requested prior to acceptance, as described below.]

Thank you for resubmitting your work entitled *HMMR* acts in the PLK1-dependent spindle positioning pathway and supports neural development" for further consideration at *eLife*. Your revised article has been favorably evaluated by Anna Akhmanova (Senior editor), a Reviewing editor, and three reviewers.

The manuscript has been much improved, but there are some remaining issues that need to be addressed before acceptance, as outlined below:

All three reviewers expressed an appreciation for the changes to the manuscript, including the addition of a variety of new experiments, the removal of some unnecessary or distracting data, and changes to the text. However, there are some issues that were not resolved to the reviewer's satisfaction, and other concerns that arose due to the new changes to the paper. For submitting a revised version, please focus on the following key points:

1) Rework of the text to provide a coherent model for HMMR function. For the first submission of the paper, the primary focus was on HMMR playing a role together with Ran at the spindle poles. Due to the new data presented in the revised version, the emphasis changed to propose a shared pathway for HMMR and Plk1. This is a quite significant change, and although you have changed the title, this has not yet achieved an integrated model or narrative in other sections of the paper. It would be very helpful to provide clarity on how you perceive the role and function of HMMR in this context throughout the paper. Although much of this could likely be addressed through changes to the text, experiments distinguishing between a function together with Ran vs. a function with Plk1 would further strengthen the paper.

2) Use appropriate caution in the interpretation of the presented experiments. There are several points at which you make strong and definitive statements that appear to go beyond the nature of the presented data. This includes the co-IP Western blotting experiments, the Ran/RanBP1 binding assays, and the Ran centrosome localization (as examples). Balancing your interpretation and conclusions may also help with point 1 above for creating an appropriate overall model.

3) An additional point that arose during the reviewer discussion was the removal of the data for the localization of GFP fusions to HMMR wt and mutant forms rather than providing improved images (localization was not clear in the prior version). If this is robust and reproducible, it would be good to include these images instead of referencing a prior paper. If this is not feasible, please provide an explanation.

The reviewer comments are included below for your reference and for additional suggestions.

*Reviewer #1:*

This paper has a couple of key appeals to me. First, it uses both genetic perturbations in mouse models and cell biology in human tissue culture cells to investigate spindle positioning in diverse contexts. Second, by analyzing the role of HMMR in the established spindle positioning pathways (including dynein/NuMA, Plk1, and Ran-based mechanisms), this can help provide insights into the overall control of this process. For this revised version, the authors have conducted substantial additional experimentation, and have also made a number of changes to the text and figures (for example, removing the *Drosophila* data as requested by the reviewers). There are some nice additions to the paper, including the analysis of Plk1. Although this work represents a step forward, I am left with the feeling that this story is not fully definitive or comprehensive. The overall take home messages have changed substantially since the first submission, necessitating a change to the title and some of the core findings. In this case, instead of proposing that HMMR is acting through Ran, now the authors are proposing Plk1 as a primary factor. At the same time, there is still confounding data arguing for an interaction with Ran (through RanBP1), but this is ill-defined. Although some of the individual datapoints are intriguing, I think that they are far away from being able to make definitive statements like "Thus, PLK1 phosphorylation of HMMR (pHMMR) enables an interaction with RanBP2." or "PLK1 activity and HMMR, which binds Ran-GTP, are required to locate constitutively active Ran at mitotic centrosomes."

The nature of the advance and such definitive statements is further complicated by the fact that many of the data and images show relatively modest effects (such as the change in the Ran-GTP interaction with RanBP1 when HMMR is depleted – looks like a blotting issue – or the effect on RanCA localization in Figure 4/G and Ran in Figure 7 – quantification looks like a modest change at best). Similarly, I don't understand mechanistically why HMMR depletion would lead to a reduction of DYNLL1 or dynactin with DHC. These type of co-IP Western experiments are quite sensitive and need to be carefully controlled. In this case, there is not a realistic explanation for this effect. The effect in cells looks more like a secondary effect on spindle structure to me rather than a direct effect on dynein interactions (they don't show dynein spindle localization, for example).

As much as I like the overall concepts and ideas in this paper, I am concerned that the substantive advances in this paper will be compromised by distracting data that does not rise to this level and that their model and conclusions currently being complicated by trying to include all of this regardless of significance. I would also strongly urge the authors to reconsider their overall model and to use caution in making strong and incontrovertible statements.

*Reviewer #2:*

The new version of the manuscript is much improved. The new focus on the Plk1/HMMR/Ran axis is much clearer, and the paper, now stripped from too much "satellite" data, reads much better.

*Reviewer #3:*

The revised manuscript by Connell, et al. largely resolves the concerns raised with the initial submission. I would now recommend for publication.

---

## [Author Response]

Essential revisions:The reviewers are in agreement that this paper is interesting, but feel that there are several things that are necessary to improve the paper to the point where it would be acceptable for publication in eLife. The primary suggestions relate to strengthening the mechanistic aspects of the work.1) Your paper suggests an important role for spindle pole-localized HMMR in signaling to the cell cortex to promote proper spindle positioning and regulating DHC and NuMA localization. Prior work has suggested that Plk1 also acts as a spindle-localized signal to control spindle position. Does this suggest that HMMR impacts Plk1 activity as well? It would be very useful to consider the relationship between these two pathways and compare the effects from individually or simultaneously targeting these pathways:• It is important test whether HMMR affects Plk1 signaling. The reviewers recommend that you combine Plk1 inhibition or cortically tethering with HMMR perturbations for the experiments in Figure 5 and 6.

HMMR does impact PLK1 activity. In the Introduction for the revised manuscript, we have referenced published studies linking HMMR with PLK1 activity, including that silencing HMMR reduces PLK1 activity at kinetochores during spindle assembly (Chen et al., Cell Cycle 2014) and that HMMR-T703 is a substrate for PLK1 as identified in a mitotic phosphoproteome (Nousiainen et al. PNAS, 2006).

These published findings, and the reviewer’s recommendations, founded our new MS analysis of proteins immunoprecipitated with antibodies against phosphorylated HMMR-T703 (pHMMR), which discovered RANBP2 (Figure 4; Figure 4—figure supplement 1; subsection “HMMR interacts with RanBP2 and regulates the centrosome localization of Ran”, first paragraph). We have performed multiple new experiments and we found that silencing HMMR dampens the intensity of phosphorylated PLK1 at spindle poles (Figure 6) and that small-molecule inhibition of PLK1 results in similar phenotypes as observed in HMMR-silenced cells, including a reduction in Ran co-localization with TUBG1 (Figure 4), reduced astral microtubule density (Figure 6), and reduced cortical NuMA localization (Figure 6). We tested if the concurrent inhibition of HMMR and PLK1 significantly altered the spindle positioning pathway when compared to the inhibition of either alone and found that the localization of NuMA was not additively altered in dual inhibited cells (Figure 6). We now conclude that HMMR acts in the PLK1-dependent spindle positioning pathway, and have modified the title of our article accordingly.

• The authors show that HMMR centrosome localization domain is important for its function in spindle orientation. This is a nice part of the study, and the authors use many different techniques to decipher this mechanism. However, as they admit in the Discussion, the relative importance of HMMR-Ran-centrosomal pathway with respect to others involving Plk1 and DYNLL1 is not determined; making less clear what is the relevance of these results. The authors may want to dissect whether Ran centrosome localization alone is important for spindle orientation by modulating centrosomal Ran levels independently of HMMR and by altering DYNLL1 and PlK1 expression and activity in the HMMR overexpression experiments.

We have performed new experiments to dissect the importance of the HMMR-Ran-centrosomal pathway with respect to others involving PLK1 or DYNLL1. Our new pHMMR IP/MS data discovered RANBP2 as a putative interaction that potentially placed the HMMR-RANBP2-Ran localization downstream of PLK1 activity. Consistently, the deficits in spindle positioning that followed PLK1 inhibition are similar to those that follow HMMR silencing and concurrent disruption of both did not have an additive effect. As outlined in our first response to comment 1 (above) and as we now summarize in Figure 7—figure supplement 2, we believe HMMR acts with PLK1 to locate Ran and modulate the centrosome-localized, spindle positioning pathway.

HMMR-CHICA-DYNLL1 is proposed to strip cortical NuMA-DHC in a PLK1-independent manner however a definitive mechanism was not described (Dunsch et al., 2012). It was proposed that HMMR-CHICA anchored DYNLL1 at the spindle pole and the consequent gradient of DYNLL1 brought proximal to the cortex with an off-center spindle pole altered the composition of cortical dynein complexes by competing with dynactin, which is needed for retention on the cortex (Dunsch et al., 2012). In a set of new experiments (Figure 6), we tested two predictions derived from this putative mechanism: (1) We IP’d DHC-GFP complexes from HMMR-silenced cells and found that co-precipitated DYNLL1 was reduced, which is consistent with the putative mechanism. However, we found that co-precipitated p150^glued^ was also reduced. (2) We localized CHICA and DYNLL1 in HMMR-silenced spindles, and found their localization to spindle fibers was reduced but both were localized at spindle poles. Thus, we conclude that silencing HMMR does alter the composition of DHC-GFP complexes and reduces the incorporation of DYNLL1; however, the incorporation of p150^glued^ is also reduced and thus HMMR-DYNLL1 may not impede dynein-dynactin complex formation. HMMR also appears dispensable for DYNLL1 and CHICA localization to the spindle poles although it does impact their localization to spindle fibers. Together, these data support a conclusion that HMMR acts in the PLK1-dependent spindle positioning pathway.

2) It would be helpful to more carefully evaluate the role of Ran in this process using Ran mutants (constitutively active or dominant negative). For example, does a Ran constitutively active mutant phenocopy HMMR overexpression? Alternatively, does Ran inhibition suppress the effects of HMMR overexpression? For the work on Ran, it would provide a useful addition if you could provide data to clarify the interaction and nature of the relationship between HMMR and Ran (direct interaction, through RanBP1, does this affect Ran activity, etc.):

Results from a new experiment demonstrate that the expression of a Ran constitutively active mutant does phenocopy the effect of HMMR overexpression on the cortical localization of NuMA (Figure 8). As recommended in Essential revision 3 (see below), we have removed the work in *Drosophila* to focus the study. The removal of the *Drosophila* work also led to the removal of the observation of cortical blebbing resulting from the ectopic expression of HMMR in neuroblasts; thus, we removed our analysis of cortical blebbing in tet-HMMR cells, as well. So, we did not examine if Ran inhibition suppresses the effects of HMMR overexpression in tet-HMMR cells. In new IP- MS analysis, we discover and validate a pHMMR-RANBP2 interaction to clarify the nature of the relationship between HMMR and active Ran (Figure 4).

• siHMMR cells still deplete NuMA from the lateral cortex, suggesting this chromosomal Ran effect is functional. The authors might consider including additional siRNA conditions, such as against Rcc1 alone or in combination with HMMR, to compare HMMR to loss of activated Ran. A discussion of how HMMR discriminates potential subcellular pools of Ran as well as how it may or may not interfere with the Plk1-Dynein regulatory loop would be beneficial.

In the revised manuscript, we have discussed the role of HMMR in the PLK1-dynein regulatory loop and contrasted that role with the putative mechanism of action published concerning the HMMR-CHICA-DYNLL1 pathway (Discussion, seventh, eighth paragraphs). We have discussed the assignment of Ran in the centrosome proteome, the role of HMMR in co-localizing Ran with TUBG1, and our observation that HMMR is dispensable for the Ran-GTP gradient generated at chromosomes (Discussion, ninth paragraph). A connection between HMMR and RCC1 in the control of the generation of Ran-GTP at chromosomes is beyond the scope of this manuscript but is a provocative concept given the central role that HMMR plays in chromosome-dependent spindle assembly (Groen et al., 2004) (Joukov et al. 2006) (Chen et al., 2014) (Scrofani et al., 2015), which is RCC1-Ran-GTP dependent.

• To further establish the link between the HMMR-Ran centrosomal pathway described in vitro and in vivo observations (and to confirm that the HMMR-centrosomal Ran pathway controls spindle orientation to regulate neurogenesis), the authors should show localization of Ran to centrosomes in vivo (the fact that they study this in neuroblastoma cells is not sufficient).

In new studies, we examined spindle position, spindle fiber density, and Ran co-localization with TUBG1 in neuroblasts lining the ventricles in *Hmmr^tm1a/tm1a^*or *Hmmr^+/+^*E14.5 brains (Figure 7).

• It is also important to consider alternative explanations for the observed phenotypes beyond altered Ran signaling.- HMMR loss is shown to amplify centrosome number in MEFs (although a discussion of how this might occur is not presented). Have the authors considered that this centrosome amplification may contribute to spindle orientation defects (e.g. see Kulukian et al. PNAS 2015) independent of any HMMR-Ran connection?- To what extent is the orientation pathway described in vitro responsible for orientation defects in vivo? Polarity defects could affect spindle orientation upstream of the force localizing machinery (which is different from the mechanism the authors propose). Spindle density, mitotic defects and centrosome amplification phenotypes should be also considered when analyzing /stating the mechanism controlling spindle orientation. Are Astral MT affected? Is mitosis length increased? In spindle pictures in vivo and in ES derived rosettes, spindle density and size seem affected. The authors should characterize these features (in vivo and /or in cells). Generally it would be useful to analyze polarity markers, centrosome amplification, spindle morphology in vivo (or at least in rosettes).

We thank the reviewers for this series of excellent observations. In the revised manuscript, we discuss alternate mitotic or post mitotic processes that are disturbed in HMMR-silenced cells (usually cancer cells) as additional explanations for observed phenotypes in *Hmmr^tm1a/tm1a^*mice (Discussion, third paragraph). As discussed (Discussion, second paragraph), we focused our studies on mitotic processes, rather than non-mitotic processes, given our observations that HMMR is expressed in cyclin-positive, apically positioned r-NSCs during the process of neural differentiation (Figure 7—figure supplement 1). We now include the findings from many additional experiments to address each of the reviewer’s comments in the following ways:

- Polarity markers appear unaffected in vivo (Figure 2—figure supplement 1).

- Centrosome amplification: We assessed centrosome numbers, and the presence/absence of multipolar spindles, in WT and KO NP cells and found no significant difference (Figure 2—figure supplement 1).

- Spindle morphology in vivo: we examined spindle position, spindle length and spindle fiber density in neuroblasts lining the ventricles in *Hmmr^tm1a/tm1a^*or *Hmmr^+/+^*E14.5 brains (Figure 7).

- Astral microtubules are affected in HMMR-silenced cells (Figure 6). We were not able to resolve astral microtubule density in vivo.

- The length of mitosis is augmented in *Hmmr^tm1a/tm1a^* MEFs and HMMR-silenced cells, which is noted in the s paragraph of the Discussion as unpublished data. These unpublished data are described in a separate manuscript, entitled Chen et al. The non-motor adaptor HMMR dampens Eg5-mediated forces to preserve the kinetics and integrity of chromosome segregation (submitted, Mol Biol Cell).

3) The reviewers also recommend that you trim aspects of your analysis to focus on the novel and substantive new insights. All three reviewers expressed confusion for the diverse data and systems, some of which was not directly relevant to the overall conclusions. The authors could do a better job of integrating this data and experimental systems. The reviewers recommend removing some of the data from the work in Drosophila cells and other experiments that are not directly related to the main points of the paper, although we leave this to your discretion.• In its current version, it seems that this paper is constructed so that it can accommodate a large amount of data obtained in different systems. The authors try to squeeze together a neurodevelopmental biology study and a cell biology study, and the resulting message is confused and confusing. The paper is difficult to read, and it needs a major overhaul.• The inclusion of the Drosophila experiments seems unnecessary as there is no complementation of the Miranda phenotype (perhaps not surprising given the evolutionary difference), and they only use this to see cortical blebbing. If the authors want to include this data, they could alter this section to minimize the analysis of the complementation (perhaps a single sentence).

We have constructed a more cohesive story by removing the *Drosophila* data and the related data addressing cortical blebbing and daughter cell size in tet-Hmmr Hela cells. We have also removed many aspects of the neural rosette data. Finally, we have focused on the novel aspects of our work by contrasting our findings from our new mouse model with the findings obtained in published *Hmmr* mutant mouse models (Li et al., 2015, 2016) (Tolg et al., 2003) and recent work from *Xenopus* embryos treated with *HMMR* morpholinos (Prgaer et al., 2017).

4) The in vivo analysis with the mouse model would benefit from some additional consideration. For the revised paper, you should use caution in the interpretation of the mouse knockout phenotypes and avoiding redundancy with other recently published work to highlight your novel discoveries:• Previous work by Li et al. also generated a HMMR mouse mutant. There are a number of similarities, but also differences between these two papers. In the previous case, the authors only eliminated the C-terminal region of the protein. Here, the authors argue that the lethality of their mutant (but not the previous mouse model) is related to functions of the N-terminus. However, the prior paper also analyzed spindle orientation with apparently similar observations. It would be helpful to further clarify the differences and similarities between the two papers.

As is noted by the reviewer in the next comment, the *Hmmr^tm1a/tm1a^*mouse model is the more potent knockout for Hmmr in mice so we do not believe there is redundancy between the findings from our work and the published *Hmmr* mice. We do note the work from Li et al. (2015, 2016) in the Introduction and we discuss the similarities and differences in the Discussion. In particular, we highlight in the discussion the recently published work in *Xenopus* HMMR morphants that indicate a critical role for the N-terminus of the protein in neural development.

• Although the systems are used somewhat interchangeably, in some cases they are not used to their full potential. The power of this paper to generate a potent knockout for HMMR in mice. However, they do not use cells derived from these mice for most of the subsequent cell biological experiments. Instead, they deplete the protein by RNAi in HeLa cells, which may be both less efficient and also prone to off target defects (no rescue is shown).

We performed many studies in siHMMR-treated HeLa cells, rather than *Hmmr* KO MEFs, due to: (1) MEFs have a low mitotic rate, which makes them suboptimal for mechanistic studies; and, (2) many of the seminal observations describing the spindle positioning pathway in mammalian cells were obtained using HeLa cells making this model an optimal one for comparison with the published literature.

With respect to rescue in siHMMR-treated HeLa cells, we have previously published that the reduced kinetics of spindle assembly observed in siHMMR-treated HeLa cells can be rescued through the expression of GFP-HMMR (Chen et al., 2014), which argues against the effects observed here (using the same constructs) being the result of off-target effects. In addition, our findings describe consistent phenotypes (changes to ran localization, spindle structure and spindle position) in redundant models for HMMR-deletion (mice, Mefs, HMMR-silenced HeLa).

• The authors state that HMMR effect is "pleiotropic" and are careful not to conclude that the observed developmental defects derive strictly from spindle orientation defects. However a large part of the Introduction is based on the idea that spindle orientation dictates cell fates in the developing CNS, and somehow directs the reader towards this interpretation. This is actually a very dogmatic view: although many published studies claim a causal relationship, they only show a correlation between strong neurogenesis defects (that may be caused by defective migration, cell death, or many other factors) and often minor defects in spindle orientation. As the authors later show in different HMMR mutant contexts, there are actually many defects:- increased cell death,- probable aneuploidy,- centrosome mispositioning (described in rosettes),- centrosome overduplications (described in MEFs)- mitotic cell death (in Figure 5)which are as many possible causes of microcephaly. If all these defects occur also in vivo (which they do not document), then one cannot conclude that neurogenic and brain formation defects are specifically generated by spindle misorientation itself. Indeed, centrosome amplification has been linked to microcephaly independently of spindle orientation (Marthiens et al., 2013). Centrosome positioning defects in rosettes suggest that polarity may be affected, and indeed the authors cite defects in a-b in mammary epithelia. Polarity defects can affect the architecture of the neuroepithelium, affecting indirectly neuroepithelium structure and neurogenesis, and it would be necessary to investigate a-b cell polarity in mutant brains.

These are excellent suggestions that overlap with aspects of those in the last criticism of point 2. As outlined in our response to that criticism, we have performed additional experiments to investigate these putative effects. With specific note to polarity defects, we looked at Par3 and ZO-1 as apical markers and their localization appears to be unaffected in vivo (Figure 2—figure supplement 1).

[Editors' note: further revisions were requested prior to acceptance, as described below.]

The manuscript has been much improved, but there are some remaining issues that need to be addressed before acceptance, as outlined below:All three reviewers expressed an appreciation for the changes to the manuscript, including the addition of a variety of new experiments, the removal of some unnecessary or distracting data, and changes to the text. However, there are some issues that were not resolved to the reviewer's satisfaction, and other concerns that arose due to the new changes to the paper. For submitting a revised version, please focus on the following key points:1) Rework of the text to provide a coherent model for HMMR function. For the first submission of the paper, the primary focus was on HMMR playing a role together with Ran at the spindle poles. Due to the new data presented in the revised version, the emphasis changed to propose a shared pathway for HMMR and Plk1. This is a quite significant change, and although you have changed the title, this has not yet achieved an integrated model or narrative in other sections of the paper. It would be very helpful to provide clarity on how you perceive the role and function of HMMR in this context throughout the paper. Although much of this could likely be addressed through changes to the text, experiments distinguishing between a function together with Ran vs. a function with Plk1 would further strengthen the paper.

The Abstract, the presentation and discussion of our findings, and the model have been edited to clarify that silencing HMMR results in dampened PLK1 activity and a disturbed PLK1-dependent positioning pathway. We moved the “HMMR interacts with RanBP2 and regulates the centrosome localization of Ran”so that it follows the “HMMR enables the PLK1-dependent spindle pole positioning pathway” section to emphasize that HMMR acts through the PLK1 positioning pathway. As a putative mechanistic explanation for how silencing HMMR may dampen the PLK1-dependent spindle positioning pathway, we discuss the literature that demonstrates HMMR enables the formation of activating Aurora A-TPX2 complexes (Chen et al., Cell Cycl*e* 2014) (Vernos and colleagues, Curr Biol 2015) and that active Aurora A directly phosphorylates PLK1-T210 (Medema and colleagues, Nature 2008). We modified the model to clarify the mechanism.

The presentation and discussion of our findings and the model have been edited to clarify that pHMMR locates active Ran at centrosomes in a PLK1-dependent manner. We highlight in the concluding paragraph of our Discussion that the relative contribution of pHMMR function with Ran vs. a function with active PLK1, potentially through HMMR-mediated modulation of Aurora A, warrants further study.

2) Use appropriate caution in the interpretation of the presented experiments. There are several points at which you make strong and definitive statements that appear to go beyond the nature of the presented data. This includes the co-IP Western blotting experiments, the Ran/RanBP1 binding assays, and the Ran centrosome localization (as examples). Balancing your interpretation and conclusions may also help with point 1 above for creating an appropriate overall model.

We appreciate the significance of this criticism. In our revised manuscript, we adjusted the interpretations and conclusions drawn from our findings, especially as they relate to the manner and significance of HMMR-Ran complexes at mitotic spindle poles.

3) An additional point that arose during the reviewer discussion was the removal of the data for the localization of GFP fusions to HMMR wt and mutant forms rather than providing improved images (localization was not clear in the prior version). If this is robust and reproducible, it would be good to include these images instead of referencing a prior paper. If this is not feasible, please provide an explanation.

We added improved images to Figure 3—figure supplement 1.

The reviewer comments are included below for your reference and for additional suggestions.Reviewer #1:[…] Although this work represents a step forward, I am left with the feeling that this story is not fully definitive or comprehensive. The overall take home messages have changed substantially since the first submission, necessitating a change to the title and some of the core findings. In this case, instead of proposing that HMMR is acting through Ran, now the authors are proposing Plk1 as a primary factor. At the same time, there is still confounding data arguing for an interaction with Ran (through RanBP1), but this is ill-defined. Although some of the individual datapoints are intriguing, I think that they are far away from being able to make definitive statements like "Thus, PLK1 phosphorylation of HMMR (pHMMR) enables an interaction with RanBP2." or "PLK1 activity and HMMR, which binds Ran-GTP, are required to locate constitutively active Ran at mitotic centrosomes."

We altered these statements to more precisely interpret our presented findings.

The nature of the advance and such definitive statements is further complicated by the fact that many of the data and images show relatively modest effects (such as the change in the Ran-GTP interaction with RanBP1 when HMMR is depleted – looks like a blotting issue – or the effect on RanCA localization in Figure 4/G and Ran in Figure 7 – quantification looks like a modest change at best).

We noted the effects are modest in the text. The effects, while modest, were reproducible (not a blotting issue) and we include immunoblots from our replicate experiments (Author response image 1, corresponds to Figure 6 in revision #2).

Similarly, I don't understand mechanistically why HMMR depletion would lead to a reduction of DYNLL1 or dynactin with DHC. These type of co-IP Western experiments are quite sensitive and need to be carefully controlled. In this case, there is not a realistic explanation for this effect. The effect in cells looks more like a secondary effect on spindle structure to me rather than a direct effect on dynein interactions (they don't show dynein spindle localization, for example).

We clarified the proposed mechanism in the text. Briefly, Barr and colleagues, J Cell Biol 2012 demonstrated that a HMMR-CHICA complex at the spindle/poles contributed to the positioning pathway, potentially in a PLK1-independent manner. HMMR-CHICA was shown to recruit DYNLL1 to the spindle/ poles and proposed to create a local increase of DYNLL1 activity. Consistently, we identified DYNLL1 and CHICA in pHMMR IPs by MS analysis and we found that the DYNLL1-DHC interaction was reduced in HMMR-silenced cells.

Barr and colleagues further speculated that the local increase of DYNLL1 caused DYNLL1- dynein interactions that resulted in loss of dynein from the cell cortex by competing with dynactin: “a prediction of this model is that the DYNLL1-bound form of dynein cannot bind to dynein adaptors required for cortical targeting” and “Therefore, DYNLL1 may compete with dynactin for binding sites on the dynein intermediate chain, and thereby displace dynein complexes from the cell cortex.” A prediction of this model would be that cortical dynein is retained in HMMR-silenced cells because dynactin is incorporated and not displaced by DYNLL1. While we found the DYNLL1-DHC interaction was reduced in HMMR-silenced cells, we do not find evidence for augmented dynactin-DHC complexes. In fact, we find that dynactin-DHC complexes are reduced. We agree that these co-IP experiments are quite sensitive. For this reason, we had 2 technicians independently verify the effect (Author response image 2, corresponds to Figure 5 in rev. #2).

**Author response image 2. respfig2:** 

As much as I like the overall concepts and ideas in this paper, I am concerned that the substantive advances in this paper will be compromised by distracting data that does not rise to this level and that their model and conclusions currently being complicated by trying to include all of this regardless of significance. I would also strongly urge the authors to reconsider their overall model and to use caution in making strong and incontrovertible statements.

We have modified the model, and the text throughout, to highlight that HMMR feeds into the intrinsic PLK1 positioning pathway. We have also highlighted the fact that while we localize active Ran at spindle poles, and that its localization is reduced by the silencing of HMMR or the inhibition of PLK1, the relative contribution of active Ran at the spindle pole to the regulation of cortical NuMA localization is not yet clear.